# ELBOing Stein: Variational Bayes with Stein Mixture Inference

**Ola Rønning**
Department of Computer Science
University of Copenhagen
ola@di.ku.dk

**Eric Nalisnick**
Department of Computer Science
Johns Hopkins University
nalisnick@jhu.edu

**Christophe Ley**
Department of Mathematics
University of Luxembourg
christophe.ley@uni.lu

**Padhraic Smyth**
Department of Computer Science
University of California, Irvine
smyth@ics.uci.edu

**Thomas Hamelryck**
Departments of Computer Science / Biology
University of Copenhagen
thamelry@bio.ku.dk

## Abstract

Stein variational gradient descent (SVGD) (Liu & Wang, 2016) performs approximate Bayesian inference by representing the posterior with a set of particles. However, SVGD suffers from variance collapse, i.e. poor predictions due to underestimating uncertainty (Ba et al., 2021), even for moderately-dimensional models such as small Bayesian neural networks (BNNs). To address this issue, we generalize SVGD by letting each particle parameterize a component distribution in a mixture model. Our method, *Stein Mixture Inference* (SMI), optimizes a lower bound to the evidence (ELBO) and introduces user-specified guides parameterized by particles. SMI extends the Nonlinear SVGD framework (Wang & Liu, 2019) to the case of variational Bayes. SMI effectively avoids variance collapse, judging by a previously described test developed for this purpose, and performs well on standard data sets. In addition, SMI requires considerably fewer particles than SVGD to accurately estimate uncertainty for small BNNs. The synergistic combination of NSVGD, ELBO optimization and user-specified guides establishes a promising approach towards variational Bayesian inference in the case of tall and wide data.

## 1 Introduction

Accurate and *safe* machine learning necessitates adequate uncertainty estimation to ensure reliability in critical applications such as autonomous vehicles and medical diagnosis. Even in problems considered solved, such as protein structure prediction, better handling of uncertainty – particularly in cases of conformational heterogeneity (Gavalda-Garcia et al., 2025) – could lead to further improvements. As current deep methods are known to be overly confident in their predictions (Szegedy et al., 2014; Nguyen et al., 2015), a more principled treatment of uncertainty is necessary. Bayesian probabilistic models are attractive as they assess model uncertainty through a coherent framework of updating data-based beliefs. However, Bayesian inference for complex models is often analytically and computationally intractable. Therefore, variational Bayes methods approximate a Bayesian posterior with a tractable variational distribution (Jordan et al., 1999; Blei et al., 2017).

Particle-based inference is an attractive approach to variational Bayes because it resides as an intermediate between variational and sampled-based methods (Saeedi et al., 2017; Domke, 2017). As a hybrid method, particle-based inference combines several desirable properties: sample efficiency, deterministic updates and asymptotic unbiasedness. Primary among particle variational inference algorithms is *Stein variational gradient descent* (SVGD) Liu & Wang (2016) due to its tractable and

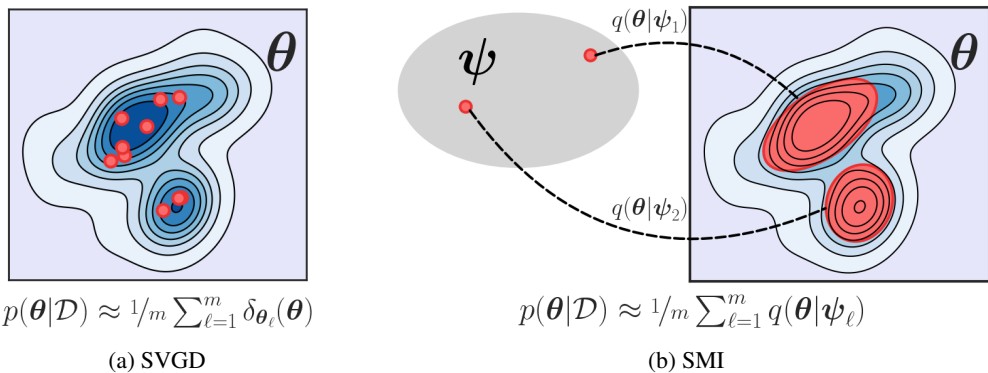

$$p(\boldsymbol{\theta}|\mathcal{D}) \approx {}^1\!/_m \sum_{\ell=1}^{m} \delta_{\boldsymbol{\theta}_\ell}(\boldsymbol{\theta})$$

(a) SVGD

$$p(\boldsymbol{\theta}|\mathcal{D}) \approx {}^1\!/_m \sum_{\ell=1}^{m} q(\boldsymbol{\theta}|\boldsymbol{\psi}_\ell)$$

(b) SMI

Figure 1: Variational inference with SVGD-derived particles (Liu & Wang, 2016) versus with an SMI-derived probability density, formulated as a mixture model (this work). **Left:** SVGD uses $m$ particles $\boldsymbol{\theta}_\ell$ to approximate the posterior $p(\boldsymbol{\theta}|\mathcal{D})$. **Right:** SMI uses a mixture model (with uniform weights) of $m$ guides $q(\boldsymbol{\theta}|\boldsymbol{\psi}_\ell)$, parameterized by particles $\boldsymbol{\psi}_\ell$ to approximate $p(\boldsymbol{\theta}|\mathcal{D})$. As a result, SMI approximates a Bayesian posterior with a richer model that alleviates variance collapse in higher dimensional posteriors.

simple update rule. However, SVGD suffers from underestimating variance, also called *variance collapse* (Ba et al., 2021; Zhuo et al., 2018). Overcoming the collapse with SVGD requires using more particles as the model size grows. We will demonstrate that this quickly becomes computationally infeasible with off-the-shelf hardware, even for moderately sized models such as small BNNs.

To address the issue of variance collapse in SVGD, we introduce *Stein mixture inference* (SMI)[1]. SMI lets each particle parameterize a component distribution, which we call a *guide*, resulting in a mixture approximation of the posterior. In contrast, SVGD directly represents approximate samples from the posterior using its particles. Figure 1 schematically distinguishes the two methods. The mixture approximation allows SMI to represent neighborhoods of SVGD particles, thereby scaling better with model size. We show that SMI is a *novel variant of Nonlinear-SVGD* (NSVGD) (Wang et al., 2019) applied to the variational approximation of Bayesian posteriors. SMI combines ordinary mean-field variational inference (OVI) (Jordan et al., 1999; Hoffman et al., 2013; Ranganath et al., 2014) with SVGD through the NSVGD framework. Specifically, our article makes the following three contributions:

1. We introduce SMI and show that it extends NSVGD to variational Bayes.

2. We empirically demonstrate that SMI is more particle efficient than SVGD.

3. We use synthetic and real-world data to show that SMI does *not* suffer from variance collapse in small- to moderately-sized models such as small BNNs.

Next, in Section 2, we will motivate SMI by outlining the reasoning behind the method.

## 2 STEIN MIXTURE INFERENCE IN A NUTSHELL

We aim to construct a richer variational approximation $q(\boldsymbol{\theta})$ of the posterior $p(\boldsymbol{\theta}|\mathcal{D})$ than the one offered by SVGD, while ensuring we also have a means to optimize it. To achieve this, we express $q(\boldsymbol{\theta})$ as a uniform mixture model of $m$ (user-defined) guides, parameterized by $m$ particles $\{\boldsymbol{\psi}_i\}_{i=1}^{m}$ that make up the *empirical measure* $\rho_m(\cdot) = {}^1\!/\sum_{i=1}^{m}\delta_{\boldsymbol{\psi}_i}(\cdot)$,

$$q(\boldsymbol{\theta}|\rho_m) = \frac{1}{m} \sum_{\ell=1}^{m} q(\boldsymbol{\theta}|\boldsymbol{\psi}_\ell). \tag{1}$$

---

[1]This article extends our preliminary work presented in the (non-archival) workshop paper Nalisnick & Smyth (2017).

The goal is to optimize the corresponding *mixture* ELBO, which measures how well the mixture model approximates the true posterior,

$$\mathcal{L}(\rho_m) = \frac{1}{m} \sum_{\ell=1}^{m} \mathbb{E}_{q(\boldsymbol{\theta}|\boldsymbol{\psi}_\ell)} \left[ \log \frac{p(\boldsymbol{\theta}, \mathcal{D})}{q(\boldsymbol{\theta}|\rho_m)} \right] \le \log p(\mathcal{D}). \tag{2}$$

Now, the mixture ELBO can be interpreted as a *symmetric*[2] *functional* $F[\rho_m]$, mapping the particles of the empirical measure to a scalar,

$$F[\rho_m] = \mathcal{L}(\rho_m).$$

This interpretation allows us to leverage the NSVGD framework to optimize $F[\rho_m]$, along with an additional weighted entropy term[3] $\alpha \mathbb{H}[\rho_m]$ to encourage particle diversity, to find

$$\rho_m^* = \arg\max_{\rho_m} F[\rho_m] + \alpha \mathbb{H}[\rho_m] = \arg\max_{\rho_m} \frac{1}{m} \sum_{\ell=1}^{m} \mathbb{E}_{q(\boldsymbol{\theta}|\boldsymbol{\psi}_\ell)} \left[ \log \frac{p(\boldsymbol{\theta}, \mathcal{D})}{q(\boldsymbol{\theta}|\rho_m)} \right] + \alpha \mathbb{H}[\rho_m], \quad (3)$$

where $\mathbb{H}[f] = -\int f \log f$ denotes the differential entropy and $\alpha \ge 0$. As we will show, despite the inclusion of the entropy term, we still obtain a proper ELBO, $\mathcal{L}_{\text{SMI}}(\rho_m) = F[\rho_m] + \mathbb{H}[\rho_m] \le \log p(\mathcal{D})$, if we choose $\alpha = 1$. This ensures the mixture model $q(\boldsymbol{\theta}|\rho_m)$ provides a well-justified, diversified posterior approximation.

## 3 BACKGROUND

After introducing OVI, we detail NSVGD in section 3.2. We state the variational objective that NSVGD maximizes, restate the central result from Wang & Liu (2019) that allows us to move $\rho_m$ in theorem 3.1 and finally, in eq. (7), give the tractable iterative update that is the backbone of NSVGD optimization.

**Notation** Let $\mathbf{x} \sim p(\mathbf{x})$ denote a sample generated from an unknown distribution $p(\mathbf{x})$. We observe $N$ independent and identically distributed draws from $p(\mathbf{x})$ that constitute the dataset $\mathcal{D} = \{\mathbf{x}_1, \dots, \mathbf{x}_N\}$. We denote the likelihood function $p(\mathcal{D}|\boldsymbol{\theta}) = \prod_{n=1}^{N} p(\mathbf{x}_n|\boldsymbol{\theta})$ where $\boldsymbol{\theta} \in \boldsymbol{\Theta} \subseteq \mathbb{R}^d$ is a latent variable. Let $p(\boldsymbol{\theta})$ denote the prior and $p(\boldsymbol{\theta}|\mathcal{D})$ the posterior. We assume that the posterior is not analytically available except up to a constant of proportionality, i.e., $p(\boldsymbol{\theta}|\mathcal{D}) \propto p(\mathcal{D}, \boldsymbol{\theta})$. We denote the differential operator as $\nabla_\ell$ when differentiating with respect to (wrt.) the $\ell$'th (random) variable. For example, $\nabla_1 f(a, b)$ is the differential wrt. $a$. However, if this notation becomes ambiguous, we use the symbolic subscript notation, e.g. $\nabla_b f(a, b) = \nabla_2 f(a, b)$.

### 3.1 ORDINARY VARIATIONAL INFERENCE

We can approximate an intractable posterior $p(\boldsymbol{\theta}|\mathcal{D})$ with a tractable variational distribution[4] $q(\boldsymbol{\theta}|\mathcal{D}; \boldsymbol{\psi})$ by optimizing a lower bound on the evidence $p(\mathcal{D})$, the aforementioned ELBO (Jordan et al., 1999). The ELBO is given by

$$\log p(\mathcal{D}) \ge E_{q(\boldsymbol{\theta}|\mathcal{D}; \boldsymbol{\psi})}[\log p(\mathcal{D}, \boldsymbol{\theta})] + \mathbb{H}[q(\boldsymbol{\theta}|\mathcal{D}; \boldsymbol{\psi})] \equiv \mathcal{L}(\boldsymbol{\psi}). \tag{4}$$

Maximizing $\mathcal{L}(\boldsymbol{\psi})$ is equivalent to minimizing the Kullback-Leibler (KL) divergence between the approximate and intractable posterior, but importantly requires computing the joint density $p(\mathcal{D}, \boldsymbol{\theta})$ instead of the intractable conditional $p(\boldsymbol{\theta}|\mathcal{D})$. $\boldsymbol{\psi}$ is obtained from maximizing the ELBO by gradient or coordinate ascent.

### 3.2 NONLINEAR STEIN VARIATIONAL GRADIENT DESCENT

Like Markov chain Monte Carlo (MCMC) methods, particle variational methods (Frank et al., 2009; Saeedi et al., 2017) approximate samples from the posterior rather than its density. However,

---

[2] A function is symmetric if its evaluation is independent of the order of its parameters.

[3] Although the differential entropy is formally undefined for an empirical measure, within the NSVGD framework (Liu et al., 2017; Wang & Liu, 2019) $\rho_m^*$ converges weakly to $\rho^*$ for $m \to \infty$. $\mathbb{H}[\rho^*]$ is well defined.

[4] We use the notation $p(c|a)$ for conditioning on the random variable $a$ and $p(c; b)$ for a density with parameters $b$.

Table 1: NSVGD generalizes SVGD and includes DivMM and SMI (this work). Like SVGD, SMI approximates general posteriors. But where SVGD represents the posterior directly with particles $\boldsymbol{\theta}_\ell$, SMI addresses variance collapse by using a mixture model $1/m \sum_{\ell=1}^{m} q(\boldsymbol{\theta}|\boldsymbol{\psi}_\ell)$ parameterized by particles $\boldsymbol{\psi}_\ell$. On the other hand, DivMM specializes NSVGD to diversified maximum likelihood estimation for mixture models and cannot approximate general posteriors; moreover, the number of particles $m$ in DivMM is a hyperparameter of the model, unlike in SVGD and SMI, where it relates to the posterior approximation's richness.

| Method | Posterior approximation | Model |
|---|---|---|
| SVGD (Liu & Wang, 2016) | $\frac{1}{m} \sum_{\ell=1}^{m} \delta_{\boldsymbol{\theta}_\ell}(\boldsymbol{\theta})$ | $p(\mathcal{D}|\boldsymbol{\theta})\pi(\boldsymbol{\theta})$ |
| DivMM (Wang & Liu, 2019) | None | $\sum_{\ell=1}^{m} p(\mathcal{D}|\boldsymbol{\theta}_\ell)$ |
| SMI (This work) | $\frac{1}{m} \sum_{\ell=1}^{m} q(\boldsymbol{\theta}|\boldsymbol{\psi}_\ell)$ | $p(\mathcal{D}|\boldsymbol{\theta})\pi(\boldsymbol{\theta})$ |

unlike MCMC methods, the number of posterior samples is fixed a priori for particle methods. Particle methods are attractive due to their freedom from strong parametric assumptions and resulting flexibility as an approximation. We will designate the samples as "particles" to emphasize that they are not auto-correlated, as with MCMC methods. However, they retain some correlation in the non-asymptotic case (Gallego & Insua, 2018).

Wang & Liu (2019) introduces the *Nonlinear* SVGD framework which allows functional optimization under diversification constraints. Wang & Liu (2019) applies this general framework to the (constrained) maximum likelihood estimation of *diversified* mixture models (DivMM), i.e., mixture models consisting of spread-out mixture components. The NSGVD framework has previously only been applied to SVGD and this type of maximum likelihood estimation.

**The variational objective**   NSVGD iteratively moves an initially simple distribution $\rho$, such as a Gaussian, according to $T(\rho) = \rho + \epsilon\phi[\rho]$ such that the transported distribution $T(\rho)$ maximally increases a variational objective. The variational objectives for NSVGD combine a functional, like the likelihood (DivMM), the negative KL-divergence to a posterior (SVGD) or an ELBO (SMI and OVI) with a diversification constraint on $\rho$. We need the constraint to avoid mode collapse.

In its most general form, NSVGD solves the maximization given by

$$\rho^* = \arg\max_{\rho} F[\rho] + \alpha\mathbb{H}[\rho], \tag{5}$$

where $F[\rho]$ is a nonlinear functional of $\rho$, $\mathbb{H}[\rho]$ is the differential entropy and $\alpha \geq 0$ scales the contribution of the entropy. The entropy acts as a regularizer, forcing $\rho$ to distribute uniformly, promoting particle diversification and avoiding collapse to the closest mode.

**Making the optimal perturbation $\phi^*$ tractable**   Finding the steepest perturbation direction $\phi^*[\rho]$ is challenging in the general setting. This is because $\phi^*[\rho]$ requires computing the *first variation* of $F$, a functional analog to the derivative of a function, which may not exist, let alone be computationally tractable. We must weaken our optimization and add additional structure on $\rho$ and $F$ to progress toward a tractable algorithm by guaranteeing the first variation is always tractable. Theorem 2 from Wang & Liu (2019) provides this structure. First, use an empirical measure $\rho_m$ on $m$ particles $\{\theta_\ell\}_{\ell=1}^{m}$ instead of $\rho$ because with the particles evaluating wrt. $\rho_m$ is trivial. However, using $\rho_m$ for $\rho$ means we only approximate the $\rho^*$ from eq. (5) with the guarantee that the optimum $\rho_m^*$ weakly converges to $\rho^*$ when letting $m \to \infty$ (Wang & Liu, 2019; Liu et al., 2017). Second, choose $F[\rho_m]$ such that there exists a symmetric and differentiable map $f : \boldsymbol{\theta}_1, \boldsymbol{\theta}_2, \ldots, \boldsymbol{\theta}_m \mapsto F[\rho_m]$. Under these two conditions, the first variation of $F$ wrt. the $\ell$'th particle reduces to ordinary differentiation of $f(\boldsymbol{\theta}_1, \boldsymbol{\theta}_2, \ldots, \boldsymbol{\theta}_m)$ which is tractable to compute.

**The optimal perturbation in RKHS**   With the additional structure on $\rho$ and $F$, the last essential component that gives $\phi^*$ a closed form is restricting the candidate perturbations to functions in a reproducing kernel Hilbert space Liu & Wang (2016). With these components, we can restate the theorem that gives the closed form for optimal perturbation as:

**Theorem 3.1.** *The Kernelized Steepest Perturbation (Wang & Liu, 2019) Let $F[\rho] + \alpha\mathbb{H}[\rho]$ be the variational objective for a transport $T(\rho) = \rho + \epsilon\phi[\rho]$, with $\epsilon > 0$ and distribution $\rho$ with*

supp $\rho \subseteq \text{dom } f \subseteq \mathbb{R}^d$. *Let $\rho_m(\cdot) = 1/m \sum_{i=1}^{m} \delta_{\boldsymbol{\theta}_i}(\cdot)$ be the empirical measure of $m$ particles and let $f : (\boldsymbol{\theta}_1, \ldots, \boldsymbol{\theta}_m) \mapsto F[\rho_m]$ be a differentiable and symmetric function. If we choose a reproducing kernel $k$ on $\mathbb{R}^d \times \mathbb{R}^d$ with reproducing kernel Hilbert space $\mathcal{H}$ (Berlinet & Thomas-Agnan, 2011) such that $\nabla_1 k$ and $\nabla_2 k$ exist and are both continuous, then the optimal perturbation direction $\phi^* \in \mathcal{H}$ such that $\| \phi^* \|_{\mathcal{H}} \leq 1$ satisfies*

$$\phi^*(\cdot) \propto \mathbb{E}_{\boldsymbol{\theta}_i \sim \rho_m}[k(\boldsymbol{\theta}, \cdot)m\nabla_i f(\boldsymbol{\theta}_1, \ldots, \boldsymbol{\theta}_m) + \alpha\nabla_1 k(\boldsymbol{\theta}, \cdot)]. \tag{6}$$

Theorem 3.1 combines Theorem 1b and 2 from Wang & Liu (2019) and provides the closed form for the optimal perturbation direction $\phi^*$ used in SMI when replacing $f$ with $\mathcal{L}(\rho_m)$. The first and second terms that constitute $\phi^*(\cdot)$ in theorem 3.1 are commonly referred to as the *attractive* and *repulsive force*. This is because the first term pulls particles towards the nearest maximum in $F$, whereas the second keeps particles from collapsing onto each other. Finally, notice we never need to evaluate $\mathbb{H}(\rho_m)$; instead, $\phi^*$ uses the kernel gradient in the repulsive force. This sidesteps the issue of not having $\mathbb{H}(\rho_m)$ available.

**The iterative optimization algorithm**   The NSVGD iterative optimization starts with particles $\{\boldsymbol{\theta}_i \sim \rho^{(0)}\}_{i=1}^{m}$ drawn from the simple initial distribution $\rho^{(0)}$. At each iteration, every particle moves according to

$$\boldsymbol{\theta}_\ell = \boldsymbol{\theta}_\ell + \epsilon \sum_{i=1}^{m} k(\boldsymbol{\theta}_i, \boldsymbol{\theta}_\ell)\nabla_i f(\boldsymbol{\theta}_1, \boldsymbol{\theta}_2, \ldots, \boldsymbol{\theta}_m) + \frac{\alpha}{m}\nabla_1 k(\boldsymbol{\theta}_i, \boldsymbol{\theta}_\ell), \tag{7}$$

which maximally increases the change in eq. (5) by theorem 3.1. We see that SVGD is an instance of NSVGD by replacing $f$ with the expected log joint density, $f(\boldsymbol{\theta}_1, \boldsymbol{\theta}_2, \ldots, \boldsymbol{\theta}_m) = \mathbb{E}_{\boldsymbol{\theta} \sim \rho_m}[\log p(\mathcal{D}, \boldsymbol{\theta})]$. In the case of SVGD, $\nabla_i f(\boldsymbol{\theta}_1, \ldots, \boldsymbol{\theta}_i, \ldots, \boldsymbol{\theta}_m)$ reduces to the scaled score function $\nabla_1 f(\boldsymbol{\theta}_i) = 1/m\nabla_1 \log p(\boldsymbol{\theta}_i, \mathcal{D})$.

**Diversified mixture models**   In the case of DivMM, Wang & Liu (2019) applies the NSVGD framework to the functional $F[\rho] = \mathbb{E}_{\boldsymbol{x} \sim \mathcal{D}}[\log \mathbb{E}_{\boldsymbol{\theta} \sim \rho}[p(\boldsymbol{x}; \boldsymbol{\theta})]]$. DivMM needs the nonlinear form of theorem 3.1 as its functional is not linear in $\rho$. The optimization is a constrained maximum likelihood estimation of the diversified mixture model, $\sum_{\ell=1}^{m} p(\mathcal{D}|\boldsymbol{\theta}_\ell)$. Here, the number of particles $m$ is a hyperparameter of the model, whereas, for SVGD and SMI, the model is independent of $m$. In table 1, we contrast SVGD, DivMM, and SMI, which are different applications of the same NSVGD framework. In the following section, we introduce SMI and demonstrate that the NSVGD framework can be adapted to infer approximate variational posteriors of general Bayesian models.

## 4   Stein mixture inference

The key to justifying SMI is showing we can optimize the SMI ELBO $\mathcal{L}_{\text{SMI}}(\rho_m)$ given by eq. (3) using NSVGD and that it is indeed an ELBO. To this end, we first show that the mixture ELBO $\mathcal{L}(\rho_m)$ given by eq. (2) is a differential symmetric function. That means we can use theorem 3.1 to find the $\rho_m^*$ that maximizes $\mathcal{L}_{\text{SMI}}(\rho_m)$ by iterating eq. (7). Next, we show that when $\alpha = 1$ in eq. (3), the resulting quantity $\mathcal{L}_{\text{SMI}}(\rho_m)$ is indeed an ELBO, despite the addition of an entropy term to the mixture ELBO $\mathcal{L}(\rho_m)$ given by eq. (2).

**The SMI function(al) and its gradient**   The mixture ELBO $\mathcal{L}(\rho_m)$ given by eq. (2) is a symmetric function wrt. $\rho_m$ due to the outer sum. If $\mathcal{L}(\rho_m)$ is also differentiable, we have the desired mapping required to use parametric differentiation to optimize eq. (2) using NSVGD. To show $\mathcal{L}(\rho_m)$ is differentiable wrt. the $\ell$'th particle, we can compute the gradient as

$$
\begin{aligned}
m\nabla_{\boldsymbol{\psi}_\ell}\mathcal{L}(\rho_m) = {}& \mathbb{E}_{q(\boldsymbol{\theta}|\boldsymbol{\psi}_\ell)}\left[\log \frac{p(\boldsymbol{\theta}, \mathcal{D})}{\sum_j q(\boldsymbol{\theta}|\boldsymbol{\psi}_j)}\nabla_{\boldsymbol{\psi}_\ell}\log q(\boldsymbol{\theta}|\boldsymbol{\psi}_\ell)\right] \\
& - \sum_{j=1}^{m} \mathbb{E}_{q(\boldsymbol{\theta}|\boldsymbol{\psi}_i)}\left[\frac{\nabla_{\boldsymbol{\psi}_\ell} q(\boldsymbol{\theta}|\boldsymbol{\psi}_\ell)}{\sum_{j=1}^{m} q(\boldsymbol{\theta}|\boldsymbol{\psi}_j)}\right].
\end{aligned}
\tag{8}
$$

If we choose $q(\boldsymbol{\theta}|\boldsymbol{\psi}_\ell)$ to be differentiable wrt. $\boldsymbol{\psi}_\ell$, we see that $\mathcal{L}(\rho_m)$ is also differentiable wrt. $\boldsymbol{\psi}_\ell$. The complete derivation is given in the appendix.

**SMIs variational objective is an ELBO**  $\mathcal{L}(\rho_m)$ is an ELBO, but does SMI indeed maximize an ELBO? For this to be the case, $\mathcal{L}_{\mathrm{SMI}}(\rho_m) = \mathcal{L}(\rho_m) + \mathbb{H}(\rho_m)$, which includes an entropic regulariser, must also be an ELBO. We show this by generalizing $\rho_m$ to the continuous case because, as noted previously, $\mathcal{H}(\rho_m)$ is technically undefined for finite particles. First, consider the continuous version of $\mathcal{L}(\rho_m)$ given by,

$$\mathcal{L}[\rho] = \lim_{m \to \infty} \frac{1}{m} \sum_{i=1}^{m} \mathbb{E}_{q(\boldsymbol{\theta}|\boldsymbol{\psi}_\ell)} \left[ \log \frac{p(\boldsymbol{\theta}, \mathcal{D})}{q(\boldsymbol{\theta}|\rho_m)} \right] = \mathbb{E}_{\rho(\boldsymbol{\psi})} \left[ \mathbb{E}_{q(\boldsymbol{\theta}|\boldsymbol{\psi})} \left[ \log \frac{p(\boldsymbol{\theta}, \mathcal{D})}{\int q(\boldsymbol{\theta}|\boldsymbol{\psi})\rho(\boldsymbol{\psi})\mathrm{d}\boldsymbol{\psi}} \right] \right] \leq \log p(\mathcal{D}).$$

Note that $\mathcal{L}(\rho_m)$ weakly converges to $\mathcal{L}[\rho]$ for $m \to \infty$ when using NSVGD. Next, we construct an upper bound $\mathcal{L}^{\uparrow}[\rho]$ of $\mathcal{L}[\rho]$,

$$\mathcal{L}^{\uparrow}[\rho] \equiv \mathbb{E}_{\rho(\boldsymbol{\psi})} \left[ \mathbb{E}_{q(\boldsymbol{\theta}|\boldsymbol{\psi})} \left[ \log \frac{p(\boldsymbol{\theta}, \mathcal{D})}{q(\boldsymbol{\theta}|\boldsymbol{\psi})} \right] \right] \geq \mathcal{L}[\rho],$$

as we show in the appendix. Now, applying SMI's variational objective given by eq. (3) to $\mathcal{L}^{\uparrow}[\rho]$ with $\alpha = 1$ indeed results in an ELBO, as shown by

$$\mathcal{L}^{\uparrow}[\rho] + \mathbb{H}[\rho] = \mathbb{E}_{\rho(\boldsymbol{\psi})} \left[ \mathbb{E}_{q(\boldsymbol{\theta}|\boldsymbol{\psi})} \left[ \log \frac{p(\boldsymbol{\theta}, \mathcal{D})}{q(\boldsymbol{\theta}|\boldsymbol{\psi})} \right] \right] + \mathbb{H}[\rho] = \mathbb{E}_{\rho(\boldsymbol{\psi})} \left[ \mathbb{E}_{q(\boldsymbol{\theta}|\boldsymbol{\psi})} \left[ \log \frac{p(\boldsymbol{\theta}, \mathcal{D})}{q(\boldsymbol{\theta}|\boldsymbol{\psi})\rho(\boldsymbol{\psi})} \right] \right]$$

$$\leq \log \mathbb{E}_{\rho(\boldsymbol{\psi})} \left[ \mathbb{E}_{q(\boldsymbol{\theta}|\boldsymbol{\psi})} \left[ \frac{p(\boldsymbol{\theta}, \mathcal{D})}{q(\boldsymbol{\theta}|\boldsymbol{\psi})\rho(\boldsymbol{\psi})} \right] \right] = \log p(\mathcal{D}).$$

Here, the inequality comes from repeatedly applying Jensen's inequality. The final equality results from simply marginalizing. Note that the objective is not an ELBO for $\alpha \neq 1$. Finally, we can conclude that $\mathcal{L}_{\mathrm{SMI}}[\rho] = \mathcal{L}[\rho] + \mathbb{H}[\rho]$ is also an ELBO, because

$$\mathcal{L}[\rho] \leq \mathcal{L}^{\uparrow}[\rho] \implies \mathcal{L}[\rho] + \mathbb{H}[\rho] \leq \mathcal{L}^{\uparrow}[\rho] + \mathbb{H}[\rho] \leq \log p(\mathcal{D}).$$

Thus, the maximizing empirical measure obtained by NSVGD, $\rho_m^*$ weakly converges to a corresponding ELBO $\mathcal{L}_{\mathrm{SMI}}(\rho^*)$ when $m \to \infty$. Our experiments indicate that SMI is generally insensitive to $\alpha$; therefore, we recommend using $\alpha = 1$ to tie the optimization to ELBO maximization.

**The iterative optimization algorithm**  Optimization starts with the empirical measure $\rho_m^1$ on particles $\{\boldsymbol{\psi}_\ell \sim \rho^0\}_{i=1}^{m}$ drawn from a simple initial distribution $\rho^0$. We subsequently iterate the following gradient ascent-like step on the particles, which by theorem 3.1 maximize eq. (3):

$$\boldsymbol{\psi}_\ell^{t+1} = \boldsymbol{\psi}_\ell^t + \epsilon \left( \sum_{i=1}^{m} k(\boldsymbol{\psi}_i^t, \boldsymbol{\psi}_\ell^t) \nabla_{\boldsymbol{\psi}_i} \mathcal{L}(\rho_m^t) + \frac{\alpha}{m} \sum_{i=1}^{m} \nabla_1 k(\boldsymbol{\psi}_i^t, \boldsymbol{\psi}_\ell^t) \right). \tag{9}$$

We continue the optimization until we reach a fixed particle configuration. In eq. (9), $\epsilon > 0$ is the step size, $k$ is a reproducing kernel, and $\alpha \in \mathbb{R}^+$ is the hyper-parameter inherent to NSVGD. $\alpha$ controls the spread of the particles (i.e., by scaling $\mathbb{H}[\rho_m]$) and is, together with the kernel $k$ and step size, the hyper-parameters of SMI.

**Connection to SVGD, OVI and maximum a posteriori**  SVGD, OVI and maximum a posteriori (MAP) estimation are all instances of SMI. In particular, where SVGD reduces to MAP estimation when only using one particle, SMI reduces to ordinary variational inference (as in eq. (4)) in the single-particle case for an arbitrary guide $q(\boldsymbol{\theta}|\boldsymbol{\psi})$. We can also connect SMI to SVGD, and by extension MAP estimation, by choosing each guide $q(\boldsymbol{\theta}|\boldsymbol{\psi}_i)$ as the point mass, i.e., $1_{\boldsymbol{\psi}_\ell}(\boldsymbol{\theta})$. These connections place SMI as a hybrid between a sample- and a density-based method. We attribute SMI's ability to mitigate variance collapse to this hybrid nature. In appendix A, we detail how SMI can be reduced to recover SVGD and OVI.

**Library implementation**  We provide an open-source implementation (under an Apache version 2 license) of SMI, called SteinVI, in the deep probabilistic programming language NumPyro (Phan et al., 2019).

## 5 RELATED WORK

There has been a flurry of work on SVGD, but much of it has concerns that are orthogonal to ours. The SVGD algorithm itself has been extended to include second-order information (Detommaso et al., 2018), operate on Riemannian manifolds (Liu & Zhu, 2018) and forgo analytic gradients (Han & Liu, 2018). Furthermore, SVGD has been re-purposed to perform message passing (Wang et al., 2018; Zhuo et al., 2018), importance sampling (Han & Liu, 2017), generative modeling (Feng et al., 2017; Pu et al., 2017), and reinforcement learning (Liu et al., 2017). Theoretical work on SVGD has analyzed its behavior in asymptotic (Liu, 2017; Lu et al., 2019; Duncan et al., 2023) and non-asymptotic (Chen et al., 2018a; Liu & Wang, 2018; Shi & Mackey, 2022) regimes as well as in high dimensions (Zhuo et al., 2018; Ba et al., 2021).

A significant part of particle VI research explores understanding SVGD as a kernelized gradient flow Liu et al. (2017); Chewi et al. (2020). This line of research investigates the kernel and the properties of its associated function space in terms of the quality of the gradient flow approximation. These include broadening the functional regularizer Dong et al. (2022), specializing the acceleration schedule on the step size Liu et al. (2019), and alternatives to RBF kernel such as scalar kernels (Gorham & Mackey, 2017; Wang et al., 2018) and matrix variate kernels Wang et al. (2019). Where SMI focuses on the attractive force, these works focus on the repulsive force. As such, there is a significant potential for adapting this body of work to SMI.

Annealing SVGD (ASVGD) D'Angelo & Fortuin (2021a) is the only alternative that directly addresses variance collapse in SVGD with a viable method. The resampling method introduced by Ba et al. (2021) is computationally impractical for large-scale problems, and we demonstrate its bias in the appendix. Our experimental results in section 6 indicate that, unlike SMI, ASVGD follows the same collapse pattern as SVGD.

This work can also be related to work on using hierarchical variational models (HVM) (Ranganath et al., 2016) and mixture approximations (Jaakkola & Jordan, 1998; Bishop et al., 1998; Gershman et al., 2012; Salimans & Knowles, 2013; Miller et al., 2017). Unlike prior work on HVMs, Stein mixture inference does not require auxiliary models or bounds looser than an ELBO. Like SMI, Morningstar et al. (2021) introduces SIWAE, a variational objective for mixture inference. While SMI employs an entropic regularizer on particles, SIWAE uses importance weighting. As an ELBO-based method, SIWAE can readily be incorporated as SMI's attractive force, enabling kernel design within the framework. We are unaware of any work that applies SVGD (Liu & Wang, 2016) or NSVGD (Wang et al., 2019) to optimize HVMs. Pu et al. (2017) considers SVGD specialized for VAEs; their method is similar to SMI in that it introduces an encoder. However, unlike SMI, their method only applies to VAEs.

## 6 EXPERIMENTS

Because SVGD and ASVGD are prone to variance collapse (Ba et al., 2021), increasing the dimensionality of the posterior requires more particles to represent uncertainty adequately. On the other hand, SMI can adjust the distribution of the variational components, thus requiring fewer particles. We demonstrate this for small and moderately sized models on synthetic and real-world data.

All experiments are carried out on an NVIDIA Quadro RTX 6000 GPU. For clarity, we only outline the experimental settings in the following sections, leaving the details necessary for reproduction to appendix C. All experiments use our publicly available SteinVI library, and we provide the source code for the experiments at `https://github.com/aleatory-science/smi_experiments`.

### 6.1 GAUSSIAN VARIANCE ESTIMATION

Following Zhuo et al. (2018), we estimate the per-dimension variance, called the dimension marginal variance, of a standard multivariate Gaussian of increasing dimension with a fixed number of particles. Hereafter, variance refers to the dimension marginal variance. The estimated variance will tend towards zero for a method prone to collapse. The right panel of fig. 2 demonstrates variance collapse in SVGD and ASVGD with twenty particles. We see no benefit from annealing SVGD. In contrast, for SMI, when we use a single particle with a mean-field multivariate Gaussian guide (i.e., the Stein

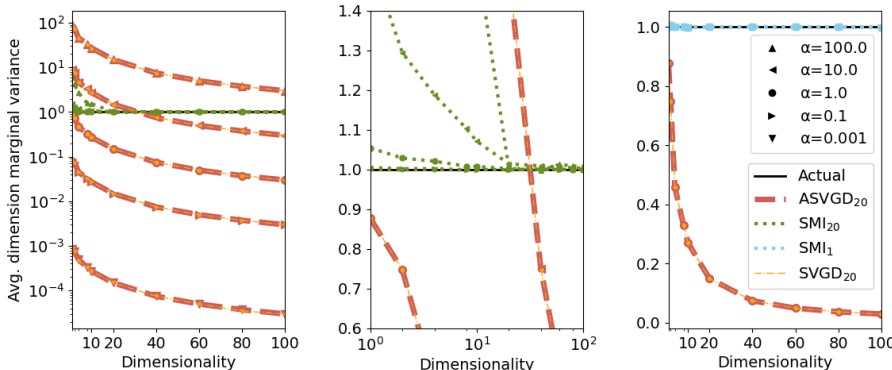

Figure 2: **Left and middle (zoom):** Variance estimation of a standard multivariate Gaussian obtained with 20-particle ASVGD, SMI and SVGD. Only SMI does not collapse and is robust to changing $\alpha$. **Right**: SMI with one particle and Gaussian guide exactly recovers the multivariate Gaussian.

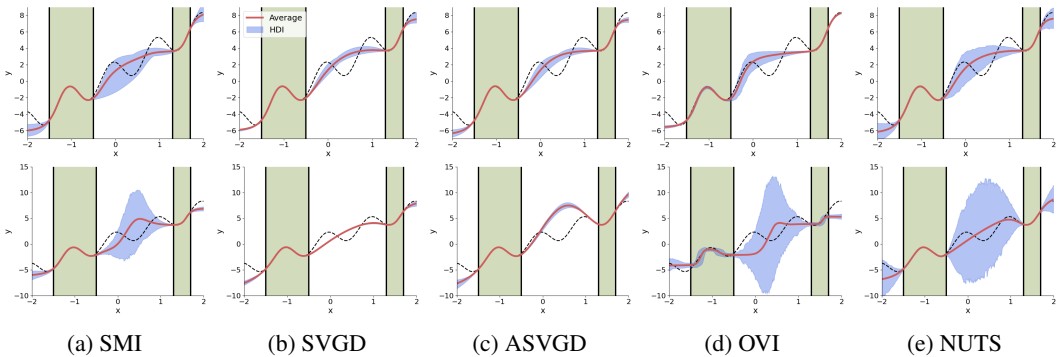

Figure 3: **Top row**: High-density interval (HDI) for the low-dimensional model inferred using SMI, SVGD, ASVGD, OVI and NUTS on the 1D wave dataset (dotted line). SVGD, ASVGD, and SMI use five particles. The posteriors are inferred with data drawn from the In region, highlighted with vertical lines. NUTS serves as a reference. **Bottom row**: HDI for the moderate-dimensional model. ASVGD and SVGD display collapse by a significant narrowing in HDI between the In regions when comparing the low to moderate dimensions. On the other hand, both OVI and SMI widen the HDI with the richer model for the in-between region. In contrast to SMI, OVI overestimates the variance in the In region, where data is available, for the mid-sized model.

particle represents the mean and variance of the guide), the estimated variance stays close to one. This is because, when using one particle, the Stein mixture contains the model.

**What happens when the SMI posterior is richer than the model?**   When the posterior is richer than the model, we risk overestimating the variance. The middle panel of fig. 2 illustrates that we can choose $\alpha \leq 1$ to improve the overestimation of variance when using more particles than needed. The model and guide are as before, but now SMI uses twenty particles instead of one. We can alleviate the overestimation with $\alpha \ll 1$ because it allows overlapping particle neighborhoods, i.e., the mixture components can collapse onto each other, thereby mimicking a single particle. The result for SMI implies that if $\alpha \ll 1$ significantly reduces variance, we are likely using too many particles. In our experiment, tuning $\alpha$ is only a viable strategy for SMI, as the right panel of fig. 2 shows. This is because $\alpha$ acts on $\rho$ for SMI, whereas choosing $\alpha \neq 1$ changes the target posterior for the other methods.

## 6.2 1D REGRESSION WITH SYNTHETIC DATA

Previously, we demonstrated that SMI with a single particle and a Gaussian guide recovers a multivariate Gaussian distribution regardless of dimension. To extend beyond this scenario, we use a synthetic one-dimensional regression dataset (dotted line in fig. 3) to study the uncertainty estimation of 2 hidden-layer BNNs in data-sparse regions. A well-calibrated model should assign high uncertainty in data-sparse areas and low uncertainty in data-rich ones (Foong et al., 2019; Daxberger et al., 2021). We compare a tiny BNN with 5 hidden units (46 random variables) to a small BNN with 100 hidden units (10,401 random variables).

**Does SMI capture uncertainty better?** Figure 3 illustrates this using the high-density interval (HDI) (Gelman et al., 1995), shown in gray, which represents the narrowest 90% Bayesian credible interval. With No-U-Turn Sampler[5] (NUTS) (Hoffman et al., 2014) serving as a reference, a well-calibrated model is expected to produce wide HDIs in data-sparse regions and narrow HDIs in data-rich areas. Among the variational methods, only SMI demonstrates this desired behavior for low- and moderate-dimensional models. Closing the SMI-SVGD gap in moderate-sized networks by increasing SVGD particles is infeasible on our hardware (appendix B). Moreover, ASVGD shows no improvement over SVGD, which is consistent with the variance experiment.

## 6.3 UCI REGRESSION BENCHMARK

To investigate the improvement in uncertainty quantification of SMI on moderately sized models for real-world data, we consider the UCI regression benchmark with Standard and Gap10 splits on 2 hidden-layered BNNs. Standard UCI uses ordinary 10% test splits (Mukhoti et al., 2018). Gap10 sorts each feature dimension to create splits (Foong et al., 2019): The middle 10% of data is used for testing, while the tails are used for training. A well-calibrated method should perform well on standard and not catastrophically deteriorate on Gap10. The BNN details, datasets and splits are summarized in appendix C.4. For comparison, we use SVGD, ASVGD, MAP and OVI as baselines and NUTS as the gold standard.

Table 2 summarizes the UCI results, evaluating their performance using root mean squared error (RMSE) and negative log-likelihood (NLL). NLL is the primary metric of interest because we evaluate uncertainty estimation. Here, SMI delivers the best performance on Standard and Gap10 UCI datasets. Notably, the RMSE is best for MAP and SMI, which means SMI has not sacrificed prediction accuracy to improve the NLL.

## 6.4 MNIST CLASSIFICATION

Next, we examine multi-class classification by applying 2 and 3 hidden-layer Bayesian Neural Networks (BNNs) to the MNIST dataset (LeCun et al., 2010). Details about the BNN configurations are provided in appendix C.3. Our evaluation includes accuracy (Acc), confidence (Conf), NLL, and several classification reliability metrics: the Brier score (Brier) (Brier, 1950), expected calibration error (ECE) (Guo et al., 2017), and maximum calibration error (MCE) (Guo et al., 2017). Among the reliability metrics, we highlight the Brier score, as ECE and MCE can be sensitive to the choice of the bin count (100 bins were used in this study).

Table 3 summarizes the results. For both BNNs, SMI generally outperforms other methods across all metrics except for ECE and MCE. Judging by the Brier score, SMI is deemed the best-calibrated method for 2 hidden-layer BNNs, while MAP and SMI exhibit comparable calibration performance in the 3-layer case. When considering all metrics collectively, SMI emerges as the preferred approach.

## 7 DISCUSSION

**Limitations** The main limitation of SMI is that the variational approximation could be misspecified by using too many particles or a poor choice of the parametric family. We saw an example of this in Section 6.1 when using twenty particles to estimate a Gaussian with SMI. Another major limitation

---

[5]While NUTS is asymptotically exact and serves as a reference, it is significantly slower to converge than variational inference methods.

Table 2: Root mean squared error (RMSE) and negative log-likelihood (NLL) for the UCI regression benchmark with standard and Gap10 splits. Lower is better for RMSE and NLL. Variational inference methods that are less than or equal in distribution to the lowest mean VI method are underlined. Similarly, the best, or equal in distribution, among all methods is highlighted in bold. We compare methods using a Mann-Whitney U (MWU) test (Mann & Whitney, 1947) at a significance level 0.05. For VI methods, SMI and MAP perform comparably in RMSE, with SMI outperforming alternatives on probabilistic calibration measured by NLL. Overall, NUTS outperformance SMI on NLL. However, of the two, only NUTS suffer severe deterioration on Gap10.

| | Standard UCI | | | | | | | | | | | |
| | NLL ($\downarrow$) | | | | | | RMSE ($\downarrow$) | | | | | |
| Dataset | SMI | SVGD | ASVGD | OVI | MAP | NUTS | SMI | SVGD | ASVGD | OVI | MAP | NUTS |
| Boston | 2.6 ± 0.6 | 7.7 ± 3.5 | 2.8 ± 0.7 | 2.6 ± 0.1 | 3.0 ± 0.8 | 2.2 ± 0.2 | 2.9 ± 0.7 | 4.1 ± 1.0 | 3.1 ± 0.9 | 4.2 ± 0.7 | 3.0 ± 0.8 | 3.6 ± 0.7 |
| Concrete | 3.4 ± 0.8 | 3.4 ± 0.3 | 4.3 ± 0.7 | 3.2 ± 0.1 | 3.9 ± 0.7 | 2.7 ± 0.3 | 4.8 ± 0.5 | 5.1 ± 0.7 | 5.0 ± 0.5 | 6.8 ± 0.4 | 4.7 ± 0.6 | 4.7 ± 0.6 |
| Energy | 0.8 ± 0.7 | 0.8 ± 0.2 | 27.1 ± 6.7 | 2.0 ± 0.0 | 1.3 ± 0.6 | 0.5 ± 1.5 | 0.4 ± 0.1 | 0.5 ± 0.1 | 1.0 ± 0.1 | 2.1 ± 0.1 | 0.4 ± 0.1 | 0.3 ± 0.1 |
| Kin8nm | −1.3 ± 0.1 | −1.2 ± 0.0 | −0.1 ± 0.0 | −1.1 ± 0.0 | −0.5 ± 0.0 | −1.4 ± 0.0 | 0.1 ± 0.0 | 0.1 ± 0.0 | 0.1 ± 0.0 | 0.1 ± 0.0 | 0.1 ± 0.0 | 0.1 ± 0.0 |
| Naval | −3.8 ± 0.4 | −0.6 ± 0.0 | −0.1 ± 0.0 | −3.4 ± 0.0 | −4.5 ± 0.1 | −3.2 ± 0.4 | 0.0 ± 0.0 | 0.0 ± 0.0 | 0.0 ± 0.0 | 0.0 ± 0.0 | 0.0 ± 0.0 | 0.0 ± 0.0 |
| Power | 2.9 ± 0.0 | 2.9 ± 0.0 | 2.9 ± 0.0 | 3.0 ± 0.1 | 2.8 ± 0.0 | 2.6 ± 0.1 | 4.2 ± 0.1 | 4.2 ± 0.1 | 4.2 ± 0.1 | 5.3 ± 0.5 | 4.2 ± 0.1 | 3.6 ± 0.2 |
| Protein | 2.7 ± 0.0 | 2.8 ± 0.0 | 3.5 ± 0.0 | 2.9 ± 0.0 | 2.8 ± 0.0 | 2.7 ± 0.0 | 4.0 ± 0.1 | 4.2 ± 0.2 | 4.9 ± 0.0 | 4.4 ± 0.0 | 4.1 ± 0.1 | 3.8 ± 0.0 |
| Wine | 1.0 ± 0.1 | 1.0 ± 0.1 | 1.1 ± 0.1 | 1.0 ± 0.0 | 1.1 ± 0.2 | 27.0 ± 18.6 | 0.7 ± 0.1 | 0.6 ± 0.0 | 0.6 ± 0.0 | 0.7 ± 0.0 | 0.6 ± 0.0 | 1.0 ± 0.1 |
| Yacht | 0.7 ± 0.2 | 1.9 ± 1.2 | 0.9 ± 0.3 | 1.4 ± 0.1 | 2.2 ± 2.0 | −0.4 ± 0.2 | 0.6 ± 0.2 | 0.7 ± 0.2 | 0.6 ± 0.2 | 1.2 ± 0.2 | 0.8 ± 0.4 | 0.7 ± 0.2 |
| | Gap10 UCI | | | | | | | | | | | |
| Boston | 5.6 ± 4.9 | 7.6 ± 5.1 | 4.1 ± 3.7 | 2.8 ± 0.6 | 5.2 ± 5.3 | 2.4 ± 0.2 | 4.9 ± 1.8 | 4.1 ± 1.8 | 4.1 ± 1.8 | 4.7 ± 1.7 | 4.1 ± 2.0 | 4.6 ± 1.4 |
| Concrete | 5.1 ± 2.7 | 8.4 ± 4.0 | 8.6 ± 3.7 | 3.5 ± 0.3 | 8.6 ± 5.4 | 3.2 ± 0.5 | 8.7 ± 3.3 | 8.2 ± 2.0 | 8.1 ± 1.9 | 2.9 ± 0.9 | 8.5 ± 3.0 | 8.7 ± 3.1 |
| Energy | 12.6 ± 19.1 | 13.2 ± 14.3 | 558.8 ± 803.8 | 2.4 ± 0.5 | 7.4 ± 10.5 | 2.1 ± 3.0 | 1.3 ± 1.0 | 1.5 ± 1.0 | 3.1 ± 2.6 | 2.9 ± 0.9 | 1.5 ± 1.3 | 0.9 ± 0.5 |
| Kin8nm | −0.5 ± 0.0 | −1.2 ± 0.1 | −0.1 ± 0.0 | −1.1 ± 0.1 | −1.1 ± 0.1 | −1.4 ± 0.1 | 0.1 ± 0.0 | 0.1 ± 0.0 | 0.1 ± 0.0 | 0.1 ± 0.0 | 0.1 ± 0.0 | 0.1 ± 0.0 |
| Naval | −3.9 ± 0.5 | −0.6 ± 0.0 | −0.1 ± 0.0 | −3.4 ± 0.1 | −4.5 ± 0.0 | 1,094.6 ± 1,690.9 | 0.0 ± 0.0 | 0.0 ± 0.0 | 0.0 ± 0.0 | 0.0 ± 0.0 | 0.0 ± 0.0 | 0.7 ± 0.7 |
| Power | 2.8 ± 0.0 | 2.8 ± 0.0 | 2.8 ± 0.0 | 3.0 ± 0.0 | 3.0 ± 0.1 | 2.8 ± 0.2 | 4.1 ± 0.1 | 4.1 ± 0.1 | 4.1 ± 0.1 | 4.7 ± 0.2 | 4.0 ± 0.2 | 4.5 ± 1.3 |
| Protein | 2.9 ± 0.1 | 3.0 ± 0.1 | 3.1 ± 0.4 | 3.0 ± 0.0 | 2.9 ± 0.1 | 2.8 ± 0.1 | 4.4 ± 0.2 | 4.5 ± 0.3 | 4.8 ± 0.2 | 4.7 ± 0.2 | 4.5 ± 0.2 | 4.3 ± 0.3 |
| Wine | 1.0 ± 0.1 | 1.0 ± 0.1 | 1.1 ± 0.1 | 1.0 ± 0.0 | 1.2 ± 0.2 | 62.8 ± 53.5 | 0.7 ± 0.1 | 0.6 ± 0.0 | 0.7 ± 0.0 | 0.7 ± 0.0 | 0.7 ± 0.1 | 1.2 ± 0.2 |
| Yacht | 2.0 ± 1.0 | 194.0 ± 109.1 | 2.1 ± 1.2 | 1.5 ± 0.1 | 78.9 ± 46.5 | 0.3 ± 0.5 | 1.2 ± 0.6 | 1.2 ± 0.5 | 1.1 ± 0.5 | 1.4 ± 0.2 | 1.3 ± 0.6 | 1.4 ± 0.9 |

Table 3: Evaluation of 2 and 3 hidden-layer BNNs for MNIST classification on several metrics: confidence (Conf), negative log-likelihood (NLL), accuracy (Acc), Brier score (Brier), expected calibration error (ECE), and maximum calibration error (MCE). Lower is better for NLL, Brier, ECE and MCE. Higher is better for Conf and Acc. All methods less than or equal in distribution to the method with the best mean are highlighted in bold. Methods are compared using an MWU test at a significance level of 0.05. Overall, SMI stands out as the preferred method.

| | 2 Hidden-Layered BNN | | | | | |
| Method | Conf ($\uparrow$) | NLL ($\downarrow$) | Acc ($\uparrow$) | Brier ($\downarrow$) | ECE ($\downarrow$) | MCE ($\downarrow$) |
| SMI | **0.979 ± 0.001** | **0.039 ± 0.003** | **0.957 ± 0.003** | **0.065 ± 0.005** | 0.148 ± 0.012 | 0.631 ± 0.047 |
| ASVGD | 0.972 ± 0.002 | 0.053 ± 0.004 | 0.949 ± 0.003 | 0.074 ± 0.005 | 0.135 ± 0.007 | 0.634 ± 0.024 |
| MAP | 0.973 ± 0.001 | 0.050 ± 0.002 | 0.952 ± 0.001 | 0.068 ± 0.000 | 0.133 ± 0.000 | **0.574 ± 0.000** |
| OVI | 0.921 ± 0.006 | 0.158 ± 0.012 | 0.908 ± 0.006 | 0.106 ± 0.007 | **0.085 ± 0.010** | 0.630 ± 0.136 |
| SVGD | 0.972 ± 0.003 | 0.054 ± 0.006 | 0.949 ± 0.004 | 0.074 ± 0.007 | 0.139 ± 0.014 | 0.653 ± 0.048 |
| | 3 Hidden-Layered BNN | | | | | |
| SMI | **0.979 ± 0.002** | **0.042 ± 0.005** | **0.956 ± 0.002** | **0.067 ± 0.003** | 0.150 ± 0.014 | 0.653 ± 0.057 |
| ASVGD | 0.956 ± 0.004 | 0.104 ± 0.011 | 0.936 ± 0.004 | 0.083 ± 0.003 | 0.132 ± 0.011 | **0.651 ± 0.075** |
| MAP | 0.976 ± 0.001 | 0.044 ± 0.003 | 0.955 ± 0.001 | **0.066 ± 0.000** | 0.126 ± 0.000 | **0.614 ± 0.000** |
| OVI | 0.913 ± 0.005 | 0.182 ± 0.012 | 0.899 ± 0.005 | 0.116 ± 0.005 | **0.084 ± 0.009** | 0.652 ± 0.133 |
| SVGD | 0.960 ± 0.004 | 0.091 ± 0.011 | 0.940 ± 0.002 | 0.081 ± 0.004 | 0.135 ± 0.013 | 0.649 ± 0.044 |

for practitioners is knowing how to choose the kernel. We present SMI using an RBF kernel, leaving a study of kernel choice to future work.

**Future directions** We present SMI as an extension to nonlinear SVGD, anchored in the kernelized gradient flows theory (Liu et al., 2017; Chewi et al., 2020). However, such flows are not necessarily the best choice of transport for mixture approximations Chen et al. (2018b); Dong et al. (2022). An open question is identifying which properties make gradient flows well-suited for mixtures.

One of the issues with Bayesian modeling of neural networks is their inherent non-identifiability, which can lead to degenerate posteriors (Yacoby et al., 2022; Roy et al., 2024). SMI provides several opportunities for addressing this issue via the choice of its kernel. Promising directions are reparameterization invariant kernels (Roy et al., 2024), probability product kernels (Jebara et al., 2004) and harnessing the connection between SMI and repulsive deep ensembles (D'Angelo & Fortuin, 2021b).

ACKNOWLEDGMENTS

We acknowledge support from the Independent Research Fund Denmark | Technology and Production Sciences (grant 9131-00025B) and the VILLUM Experiment Programme (grant 50240). We thank Ahmad Salim Al-Sibahi, Martin Jankowiak and Du Phan for their discussions and help in implementing the SMI inference engine in NumPyro.

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

## A    STEIN MIXTURE INFERENCE DETAILS

This section provides the details missing from section 4. In particular, we show that the functional $\mathcal{L}^{\uparrow}[\rho]$ is an upper bound to $\mathcal{L}[\rho]$ in appendix A.1, give the complete derivation of the gradient of $\mathcal{L}(\rho_m)$ in appendix A.2 and finally, in appendix A.3, demonstrate how to reduce SMI to SVGD and OVI.

### A.1    BOUNDING THE SMI FUNCTIONAL

Our goal is to show that

$$\mathcal{L}^{\uparrow}[\rho] \geq \mathcal{L}[\rho]$$

because it allows us to conclude that the SMI variational objective is an ELBO when the repulsion is not scaled (i.e., $\alpha = 1$). Recall that the above functionals are given by

$$\mathcal{L}[\rho] \equiv \mathbb{E}_{\rho(\boldsymbol{\psi})} \left[ \mathbb{E}_{q(\boldsymbol{\theta}|\boldsymbol{\psi})} \left[ \log \frac{p(\boldsymbol{\theta}, \mathcal{D})}{q(\boldsymbol{\theta}|\rho)} \right] \right]$$

and

$$\mathcal{L}^{\uparrow}[\rho] \equiv \mathbb{E}_{\rho(\boldsymbol{\psi})} \left[ \mathbb{E}_{q(\boldsymbol{\theta}|\boldsymbol{\psi})} \left[ \log \frac{p(\boldsymbol{\theta}, \mathcal{D})}{q(\boldsymbol{\theta}|\boldsymbol{\psi})} \right] \right] .$$

In both functionals, $\rho$ is a continuous distribution, $p(\boldsymbol{\theta}, \mathcal{D})$ is the joint distribution of latent variables and data, and $q(\boldsymbol{\theta}|\rho) = \mathbb{E}_{\rho(\boldsymbol{\psi})} \left[ q(\boldsymbol{\theta}|\boldsymbol{\psi}) \right]$ is SMI's variational approximation.

We need the following inequality to show that we can bound $\mathcal{L}[\rho]$. To shorten the notation, let $\rho = \rho(\boldsymbol{\psi})$ and $q = q(\boldsymbol{\theta}|\boldsymbol{\psi})$. With this notation, it holds that

$$\mathbb{E}_{\rho} \left[ \mathbb{E}_q \left[ \log \mathbb{E}_{\rho} \left[ q \right] \right] \right] \geq \mathbb{E}_{\rho} \left[ \mathbb{E}_q \left[ \log q \right] \right] \tag{10}$$

as shown by

$$\mathbb{E}_{\rho} \left[ \mathbb{E}_q \left[ \log \mathbb{E}_{\rho} \left[ q \right] \right] \right] \geq \mathbb{E}_{\rho} \left[ \mathbb{E}_q \left[ \mathbb{E}_{\rho} \left[ \log q \right] \right] \right] = \mathbb{E}_{\rho} \left[ \mathbb{E}_{\rho} \left[ \mathbb{E}_q \left[ \log q \right] \right] \right] = \mathbb{E}_{\rho} \left[ \mathbb{E}_q \left[ \log q \right] \right] .$$

With the inequality established, we can show that $\mathcal{L}^{\uparrow}[\rho]$ upper bounds $\mathcal{L}[\rho]$ as follows:

$$\mathcal{L}^{\uparrow}[\rho] \equiv \mathbb{E}_{\rho(\boldsymbol{\psi})} \left[ \mathbb{E}_{q(\boldsymbol{\theta}|\boldsymbol{\psi})} \left[ \log \frac{p(\boldsymbol{\theta}, \mathcal{D})}{q(\boldsymbol{\theta}|\boldsymbol{\psi})} \right] \right] \qquad \text{(expand log)}$$

$$= \mathbb{E}_{\rho(\boldsymbol{\psi})} \left[ \mathbb{E}_{q(\boldsymbol{\theta}|\boldsymbol{\psi})} \left[ \log p(\boldsymbol{\theta}, \mathcal{D}) \right] \right] - \mathbb{E}_{\rho(\boldsymbol{\psi})} \left[ \mathbb{E}_{q(\boldsymbol{\theta}|\boldsymbol{\psi})} \left[ \log q(\boldsymbol{\theta}|\boldsymbol{\psi}) \right] \right] \qquad \text{(negation of eq. (10))}$$

$$\geq \mathbb{E}_{\rho(\boldsymbol{\psi})} \left[ \mathbb{E}_{q(\boldsymbol{\theta}|\boldsymbol{\psi})} \left[ \log p(\boldsymbol{\theta}, \mathcal{D}) \right] \right] - \mathbb{E}_{\rho(\boldsymbol{\psi})} \left[ \mathbb{E}_{q(\boldsymbol{\theta}|\boldsymbol{\psi})} \left[ \log \mathbb{E}_{\rho(\boldsymbol{\psi})} \left[ q(\boldsymbol{\theta}|\boldsymbol{\psi}) \right] \right] \right] \qquad \text{(combine logs)}$$

$$= \mathbb{E}_{\rho(\boldsymbol{\psi})} \left[ \mathbb{E}_{q(\boldsymbol{\theta}|\boldsymbol{\psi})} \left[ \log \frac{p(\boldsymbol{\theta}, \mathcal{D})}{\mathbb{E}_{\rho(\boldsymbol{\psi})} \left[ q(\boldsymbol{\theta}|\boldsymbol{\psi}) \right]} \right] \right] \equiv \mathcal{L}[\rho],$$

which shows that $\mathcal{L}^{\uparrow}[\rho] \geq \mathcal{L}[\rho]$ as claimed.

### A.2    SMI GRADIENT DERIVATION

With the bound on $\mathcal{L}[\rho]$ established, we now focus on showing that the mapping $\boldsymbol{\psi}_1, \boldsymbol{\psi}_2, \ldots, \boldsymbol{\psi}_m \mapsto \mathcal{L}(\rho_m)$ is differentiable wrt. the $\ell$'th particle. For conciseness, we use $\mathcal{L}(\rho_m)$ to mean both the map $\boldsymbol{\psi}_1, \boldsymbol{\psi}_2, \ldots, \boldsymbol{\psi}_m \mapsto \mathcal{L}(\rho_m)$ from particles $\{\boldsymbol{\psi}_i\}_{i=1}^m$ and the functional $\mathcal{L}$ parameterized by $\rho_m$. We can do this because $\{\boldsymbol{\psi}_i\}_{i=1}^m$ completely characterises $\rho_m(\cdot) = 1/m \sum_{i=1}^m \delta_{\boldsymbol{\psi}_i}(\cdot)$.

Demonstrating that $\mathcal{L}(\rho_m)$ is differentiable is an essential component for using theorem 3.1 to optimize our variational objective. Specifically, theorem 3.1 requires us to have a *differentiable* symmetric mapping $\mathcal{L}(\rho_m)$. We already established the symmetric nature of $\mathcal{L}(\rho_m)$ in the main article. The closed form of the gradient shows that $\mathcal{L}(\rho_m)$ is differentiable wrt. $\boldsymbol{\psi}_\ell$ if $q(\boldsymbol{\theta}|\boldsymbol{\psi})$ is differentiable wrt. $\boldsymbol{\psi}$. In practice, this restricts us to guides $q(\boldsymbol{\theta}|\boldsymbol{\psi})$ that are differentiable. However, this restriction is shared with OVI and easy to fulfill.

Recall we claimed that the gradient of $\mathcal{L}(\rho_m)$ wrt. $\boldsymbol{\psi}_\ell$ particle is given by

$$
\begin{aligned}
\nabla_{\boldsymbol{\psi}_\ell} \mathcal{L}(\rho_m) = {} & \mathbb{E}_{q(\boldsymbol{\theta}|\boldsymbol{\psi}_\ell)} \left[ \nabla_{\boldsymbol{\psi}_\ell} \log q(\boldsymbol{\theta}|\boldsymbol{\psi}_\ell) \log \frac{p(\boldsymbol{\theta}, \mathcal{D})}{\sum_{j=1}^m q(\boldsymbol{\theta}|\boldsymbol{\psi}_j)} \right] \\
& - \sum_{i=1}^m \mathbb{E}_{q(\boldsymbol{\theta}|\boldsymbol{\psi}_i)} \left[ \frac{\nabla_{\boldsymbol{\psi}_\ell} q(\boldsymbol{\theta}|\boldsymbol{\psi}_\ell)}{\sum_{j=1}^m q(\boldsymbol{\theta}|\boldsymbol{\psi}_j)} \right],
\end{aligned}
\tag{11}
$$

with the SMI functional given by

$$
\mathcal{L}(\rho_m) = \frac{1}{m} \sum_{i=1}^m \mathbb{E}_{\boldsymbol{\theta} \sim q(\boldsymbol{\theta}|\boldsymbol{\psi}_i)} \left[ \log \frac{p(\boldsymbol{\theta}, \mathcal{D})}{\frac{1}{m} \sum_{j=1}^m q(\boldsymbol{\theta}|\boldsymbol{\psi}_j)} \right]
$$

To show that eq. (11) holds first, notice that because $\mathcal{L}(\rho_m)$ is symmetric, the ordering of the particles does not matter. For our derivation, we, therefore, simply pick one. With the ordering of the particles now fixed, we can derive the gradient as follows:

$$
\begin{aligned}
m \nabla_{\boldsymbol{\psi}_\ell} \mathcal{L}(\rho_m) = {} & \nabla_{\boldsymbol{\psi}_\ell} \sum_{i=1}^m \int q(\boldsymbol{\theta}|\boldsymbol{\psi}_i) \log \frac{p(\boldsymbol{\theta}, \mathcal{D})}{\frac{1}{m} \sum_{j=1}^m q(\boldsymbol{\theta}|\boldsymbol{\psi}_j)} d\boldsymbol{\theta} \qquad \text{(expand the log)} \\
= {} & \nabla_{\boldsymbol{\psi}_\ell} \sum_i \int q(\boldsymbol{\theta}|\boldsymbol{\psi}_i) \log p(\boldsymbol{\theta}, \mathcal{D}) d\boldsymbol{\theta} \\
& - \nabla_{\boldsymbol{\psi}_\ell} \sum_i \int q(\boldsymbol{\theta}|\boldsymbol{\psi}_i) \log \frac{1}{m} d\boldsymbol{\theta} \\
& - \nabla_{\boldsymbol{\psi}_\ell} \sum_i \int q(\boldsymbol{\theta}|\boldsymbol{\psi}_i) \log \sum_j q(\boldsymbol{\theta}|\boldsymbol{\psi}_j) d\boldsymbol{\theta}.
\end{aligned}
$$

Now, the second term is zero because

$$
\nabla_{\boldsymbol{\psi}_\ell} \sum_i \int q(\boldsymbol{\theta}|\boldsymbol{\psi}_i) \log \frac{1}{m} d\boldsymbol{\theta} = \nabla_{\boldsymbol{\psi}_\ell} \log \frac{1}{m} \int q(\boldsymbol{\theta}|\boldsymbol{\psi}_\ell) d\boldsymbol{\theta} = \nabla_{\boldsymbol{\psi}_\ell} \log \frac{1}{m} = 0,
$$

which gives us

$$
m \nabla_{\boldsymbol{\psi}_\ell} \mathcal{L}(\rho_m) = \nabla_{\boldsymbol{\psi}_\ell} \sum_i \int q(\boldsymbol{\theta}|\boldsymbol{\psi}_i) \log p(\boldsymbol{\theta}, \mathcal{D}) d\boldsymbol{\theta} - \nabla_{\boldsymbol{\psi}_\ell} \sum_i \int q(\boldsymbol{\theta}|\boldsymbol{\psi}_i) \log \sum_j q(\boldsymbol{\theta}|\boldsymbol{\psi}_j) d\boldsymbol{\theta}.
$$

Noting that when $i \neq \ell$ we have $\nabla_{\boldsymbol{\psi}_\ell} q(\boldsymbol{\theta}|\boldsymbol{\psi}_i) = 0$, we can eliminate the sum on the first term to have

$$
m \nabla_{\boldsymbol{\psi}_\ell} \mathcal{L}(\rho_m) = \int \nabla_{\boldsymbol{\psi}_\ell} q(\boldsymbol{\theta}|\boldsymbol{\psi}_\ell) \log p(\boldsymbol{\theta}, \mathcal{D}) d\boldsymbol{\theta} - \nabla_{\boldsymbol{\psi}_\ell} \sum_i \int q(\boldsymbol{\theta}|\boldsymbol{\psi}_i) \log \sum_j q(\boldsymbol{\theta}|\boldsymbol{\psi}_j) d\boldsymbol{\theta}.
$$

From here, if we use the product rule and combine like terms, we obtain

$$
m \nabla_{\boldsymbol{\psi}_\ell} \mathcal{L}(\rho_m) = \int \nabla_{\boldsymbol{\psi}_\ell} q(\boldsymbol{\theta}|\boldsymbol{\psi}_\ell) \log \frac{p(\boldsymbol{\theta}, \mathcal{D})}{\sum_j q(\boldsymbol{\theta}|\boldsymbol{\psi}_j)} d\boldsymbol{\theta} - \sum_i \int q(\boldsymbol{\theta}|\boldsymbol{\psi}_i) \nabla_{\boldsymbol{\psi}_\ell} \log \sum_j q(\boldsymbol{\theta}|\boldsymbol{\psi}_j) d\boldsymbol{\theta}.
$$

Finally, because $\nabla \log f = \frac{1}{f} \nabla f$ we have

$$
\begin{aligned}
m \nabla_{\boldsymbol{\psi}_\ell} \mathcal{L}(\rho_m) = {} & \int \nabla_{\boldsymbol{\psi}_\ell} q(\boldsymbol{\theta}|\boldsymbol{\psi}_\ell) \log \frac{p(\boldsymbol{\theta}, \mathcal{D})}{\sum_j q(\boldsymbol{\theta}|\boldsymbol{\psi}_j)} d\boldsymbol{\theta} - \sum_i \int q(\boldsymbol{\theta}|\boldsymbol{\psi}_i) \frac{\nabla_{\boldsymbol{\psi}_\ell} q(\boldsymbol{\theta}|\boldsymbol{\psi}_\ell)}{\sum_j q(\boldsymbol{\theta}|\boldsymbol{\psi}_j)} d\boldsymbol{\theta} \\
= {} & \mathbb{E}_{q(\boldsymbol{\theta}|\boldsymbol{\psi}_\ell)} \left[ \nabla_{\boldsymbol{\psi}_\ell} \log q(\boldsymbol{\theta}|\boldsymbol{\psi}_\ell) \log \frac{p(\boldsymbol{\theta}, \mathcal{D})}{\sum_{j=1}^m q(\boldsymbol{\theta}|\boldsymbol{\psi}_j)} \right] - \sum_{i=1}^m \mathbb{E}_{q(\boldsymbol{\theta}|\boldsymbol{\psi}_i)} \left[ \frac{\nabla_{\boldsymbol{\psi}_\ell} q(\boldsymbol{\theta}|\boldsymbol{\psi}_\ell)}{\sum_{j=1}^m q(\boldsymbol{\theta}|\boldsymbol{\psi}_j)} \right].
\end{aligned}
$$

From the above, we have established that eq. (11) holds and $\mathcal{L}(\rho_m)$ is therefore differentiable and symmetric as required for using theorem 3.1 to maximize SMI's variational objective.

### A.3 Reducing SMI to OVI and SVGD

In the following, we establish that both OVI and SVGD are instances of SMI for a particular choice of hyper-parameters, namely a single particle and a point-mass guide, respectively.

#### A.3.1 SVGD and MAP are special cases of SMI

We can connect SMI to SVGD by choosing each guide component $q(\boldsymbol{\theta}|\boldsymbol{\psi}_i)$ as a point-mass, i.e., $1_{\boldsymbol{\psi}_i}(\boldsymbol{\theta})$. Subsequently, the point-mass can be interpreted as a simple variable renaming. Using the point-mass for each particle, we have that

$$\int 1_{\boldsymbol{\psi}_i}(\boldsymbol{\theta}) \log \frac{p(\boldsymbol{\theta}, \mathcal{D})}{\frac{1}{m} 1_{\boldsymbol{\psi}}(\boldsymbol{\theta})} d\boldsymbol{\theta} = \log \frac{p(\boldsymbol{\psi}_i, \mathcal{D})}{\frac{1}{m}}.$$

Substituting this into $\mathcal{L}(\rho_m)$, the gradient wrt. the $\ell$'th particle becomes

$$\nabla_{\boldsymbol{\psi}_\ell} \mathcal{L}(\rho_m) = \nabla_{\boldsymbol{\psi}_\ell} \sum_i \log \frac{p(\boldsymbol{\psi}_i, \mathcal{D})}{\frac{1}{m}} = \nabla_{\boldsymbol{\psi}_\ell} \log p(\boldsymbol{\psi}_\ell, \mathcal{D}). \tag{12}$$

Substituting eq. (12) for eq. (2) in eq. (9) recovers the SVGD update rule given to a constant factor $1/m$. From the connection to SVGD, we get the connection to MAP estimation for free as it corresponds to SVGD with one particle (Liu & Wang, 2016). To be precise, MAP estimation corresponds to SMI with a point-mass guide and one particle. Naturally, we can also recover MAP estimation by first considering one particle and then introducing the point-mass guide. Next, we demonstrate that if we choose an arbitrary (differential) guide and one particle, then SMI corresponds to OVI.

#### A.3.2 Reducing SMI to OVI

Like SVGD reduces to MAP estimation when only using one particle, SMI reduces to ordinary variational inference (as in eq. (4)) in the single-particle case. To see this, first note that with one particle, the kernel $k(\boldsymbol{\psi}, \boldsymbol{\psi})$ is constant, regardless of $\boldsymbol{\psi}$, and thus $\nabla_1 k(\boldsymbol{\psi}, \boldsymbol{\psi}) = 0$. Starting from eq. (9) and denoting the constant value of $k(\boldsymbol{\psi}^t, \boldsymbol{\psi}^t)$ by $c$, we obtain

$$\begin{aligned}
\boldsymbol{\psi}^{t+1} &= \boldsymbol{\psi}^t + \epsilon k(\boldsymbol{\psi}^t, \boldsymbol{\psi}^t) \nabla_{\boldsymbol{\psi}} \mathcal{L}(\rho_1^t) + \epsilon \alpha \nabla_1 k(\boldsymbol{\psi}^t, \boldsymbol{\psi}^t) \\
&= \boldsymbol{\psi}^t + \epsilon c \mathbb{E}_{q(\boldsymbol{\theta}|\boldsymbol{\psi}^t)} \left[ \nabla_{\boldsymbol{\psi}}^1 \log q(\boldsymbol{\theta}|\boldsymbol{\psi}) \log \frac{p(\boldsymbol{\theta}, \mathcal{D})}{q(\boldsymbol{\theta}|\boldsymbol{\psi})} \right] - \epsilon c \mathbb{E}_{q(\boldsymbol{\theta}|\boldsymbol{\psi}^t)} [\nabla_{\boldsymbol{\psi}} \log q(\boldsymbol{\theta}|\boldsymbol{\psi})] \\
&= \boldsymbol{\psi}^t + \epsilon c \int \nabla_{\boldsymbol{\psi}} q(\boldsymbol{\theta}|\boldsymbol{\psi}) \log \frac{p(\boldsymbol{\theta}, \mathcal{D})}{q(\boldsymbol{\theta}|\boldsymbol{\psi})} d\boldsymbol{\theta} - \epsilon c \mathbb{E}_{q(\boldsymbol{\theta}|\boldsymbol{\psi}^t)} [\nabla_{\boldsymbol{\psi}} \log q(\boldsymbol{\theta}|\boldsymbol{\psi})] \\
&= \boldsymbol{\psi}^t + \epsilon c \nabla_{\boldsymbol{\psi}} \mathbb{E}_{q(\boldsymbol{\theta}|\boldsymbol{\psi}^t)} \left[ \log \frac{p(\boldsymbol{\theta}, \mathcal{D})}{q(\boldsymbol{\theta}|\boldsymbol{\psi})} \right] = \boldsymbol{\psi}^t + \epsilon c \nabla_{\boldsymbol{\psi}} \mathcal{L}(\boldsymbol{\psi}),
\end{aligned}$$

where $\epsilon > 0$ is the step size. This means that with one particle, we are doing gradient ascent on the ELBO as defined in eq. (4). The connections to SVGD and ordinary VI are attractive because SMI thus naturally bridges particle methods and OVI.

### A.4 Mini-batching

As with SVGD, computing the likelihood can become prohibitively expensive for large data sets ($N \gg 1$). To avoid the computational dependence on the size of the dataset, we approximate the likelihood by data subsampling with the unbiased estimator

$$p_{\mathcal{I}}(\mathcal{D}|\boldsymbol{\theta}) = \prod_{i \in \mathcal{I}} p(\mathcal{D}_i|\boldsymbol{\theta})^{N/|\mathcal{I}|}, \tag{13}$$

where $\mathcal{I} \subset \pi(\mathcal{D})$ and $\pi$ is a draw from the uniform distribution over index permutations. This follows the standard mini-batching method in NumPyro (Phan et al., 2019).

Table 4: The median recovery point R ($>$ 5 favors SMI) for BNNs inferred with SMI and SVGD on different regions of the wave dataset. SMI uses five particles. Due to reaching hardware limitations, the moderate-dimensional results are lower bounds.

| Model size | In | Between | Entire |
|---|---|---|---|
| Low | 1 | 8 | 8 |
| Moderate | 1 | $> 256$ | $> 256$ |

## B  RECOVERY POINT EXPERIMENT

To quantify the difference visualized in fig. 3, we use the *log point-wise predictive density* (LPPD), a quantity used for model comparison and model fit in the presence of outliers (Vehtari et al., 2017). The empirical LPPD is given by

$$\text{LPPD} = \sum_{i=1}^{n} \log \left( \frac{1}{S} \sum_{s=1}^{S} p(y_i | x_i, \boldsymbol{\theta}_s) \right),$$

where $\{(x_i, y_i)\}_i$ are data points from an evaluation region, $\boldsymbol{\theta}_s \sim q(\boldsymbol{\theta} | \boldsymbol{\psi}_i, \mathcal{D})$ and $\boldsymbol{\psi}_i$ is drawn uniformly from the converged particles. We repeat the experiment ten times to estimate the empirical LPPD for each region.

A recovery-point experiment compares the LPPD from SMI using five particles to SVGD with an increasing number of particles. We call the number of particles such that SVGD produces a better LPPD the *recovery point*, $R$. Table 4 reports the median $R$ over ten repeated trials over the three regions from Table 5.

**Can SVGD become on par with SMI by increasing the number of particles?**   Table 4 shows that increasing the particle count can only compensate for the difference in LPPD between SMI and SVGD for the tiny BNN, the low-dimensional model. For the moderately dimensional model, SVGD reaches the GPU memory limit before reaching the recovery point. The In region results show that a MAP estimate is enough only when the noise level is low and there is enough data.

## C  EXPERIMENTAL DETAILS

This section provides extra results and the experimental setup needed for reproduction. Our experimental code is available at `https://github.com/aleatory-science/smi_experiments`. SMI is available as an inference engine under the Apache V2 license as part of the deep probabilistic programming language NumPyro (Phan et al., 2019).

### C.1  VARIANCE ESTIMATION

In this experiment, we aim to recover the per-dimension variance of a multivariate standard Gaussian (MVG) across increasing dimensions. Specifically, we evaluate MVGs with dimensions 1, 2, 4, 8, 10, 20, 40, 60, 80, and 100. We compare the performance of SVGD and ASVGD, each utilizing 20 particles, against SMI with both 1 and 20 particles in estimating the variance.

For SMI, we employ a factorized Gaussian guide initialized with a scale of 0.1. SMI's Gaussian guide mean is uniformly initialized within each dimension's $[-2, 2]$. In contrast, the particles for ASVGD and SVGD are uniformly initialized within $[-20, 20]$. This wider initialization is crucial, as ASVGD and SVGD fail to converge in lower dimensions without it.

Optimization is performed using the Adam optimizer for SVGD and ASVGD and Adagrad for SMI, each with a learning rate of $0.05$. We run the optimization for 60,000 steps, sufficient for all three methods to achieve convergence.

**Posterior shape**   To assess each method's ability to recover the shape of the standard MVG, we calculate the Frobenius norm between the estimated covariance matrix and the identity matrix,

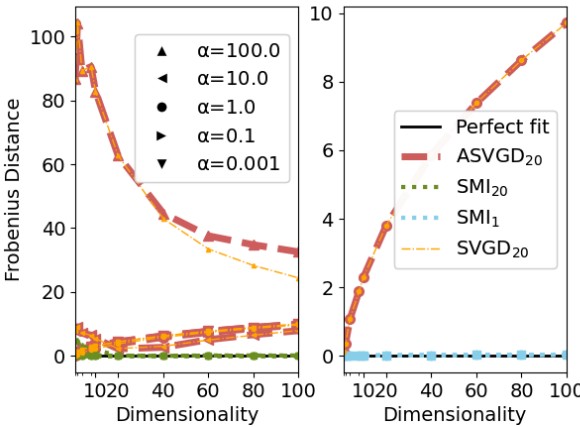

Figure 4: **Left**: Frobenius distance between the estimated and the true covariance matrix in the Gaussian variance estimation experiment, using 20 particles for all methods. Only SMI achieves distances close to zero, indicating that it accurately captures the shape of the standard Gaussian, unlike the other methods. **Right**: Frobenius distance when SMI uses a single particle. In this case, SMI perfectly recovers the posterior.

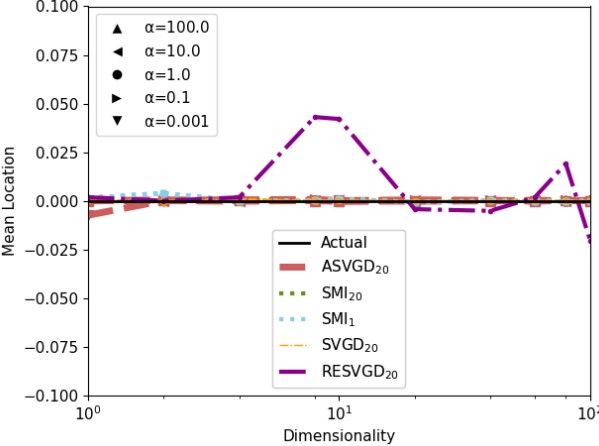

Figure 5: Mean location estimates of a standard Gaussian distribution across different dimensionalities and repulsion scaling ($\alpha$) for SMI (with 1 and 20 particles), ASVGD, SVGD and RESVGD (with 20 particles). RESVGD repulsion is not scaled. The "Actual" line represents the true mean location (zero). Only RESVGD exhibits significant bias, particularly in higher dimensions.

representing the true covariance of the MVG. A perfect recovery corresponds to a zero distance between the matrices. As illustrated in Figure 4, SMI is the only method among the three that successfully captures the shape of the standard MVG.

**Estimation of Mean Location**   Our Gaussian variance estimation experiment reveals that SVGD and ASVGD suffer from variance collapse. However, providing unbiased estimates of the mean location of a standard Gaussian distribution is an equally important requirement for these methods. As shown in fig. 5, SMI, SVGD, and ASVGD successfully achieve this. In contrast, fig. 5 also demonstrates that SVGD with resampling (RESVGD), implemented as Algorithm 1 in Ba et al. (2021), produces a biased estimate of the mean. For this reason, we excluded it from our experiments.

Table 5: Evaluation interval and data size ($|\mathcal{D}|$) of wave datasets. All data points are drawn uniformly from the evaluation interval. The Between and Entire regions contain points outside the clusters used for inference.

| Region | Evaluation Interval | $|\mathcal{D}|$ |
|--------|---------------------|-----------------|
| In | $[-1.5, -0.5] \cup [1.3, 1.7]$ | 20 |
| Between | $[-0.5, 1.3]$ | 60 |
| Entire | $[-2, 2]$ | 120 |

## C.2 SYNTHETIC 1D REGRESSION

The data-generating process combines a linear and sine-wave periodic trend given by

$$p(y|x) = \mathcal{N}\left(y|\mu = (1.5\sin\left[2\pi(x + {}^2/_3)\right] + 3x + 1\right), \sigma = 0.1\right).$$

We estimate a tiny and a small BNN using twenty observations drawn uniformly from each of two separate clusters at the intervals $[-1.5, -0.5]$ and $[1.3, 1.7]$. The construction provides a data-sparse interval $[-0.5, 1.3]$ in between the two clusters. The idea of Foong et al. (2019) is to use this in-between region to evaluate the inference methods' ability to capture and assign high uncertainty to data-sparse intervals.

We evaluate the BNNs on the In, Between, and Entire regions specified in table 5. The Between and Entire regions contain points outside the data clusters as seen in fig. 6. The In region has separate samples for inference and evaluation.

**Bayesian networks**  The BNNs have two hidden layers with `tanh` activation for both models. The moderate-dimensional case has a hidden dimension of 100, and the low-dimensional one has a hidden dimension of 5, yielding 10,401 and 46 parameters, respectively. We use standard Gaussian priors on weights and biases, and a Gaussian likelihood. The network determines the likelihood mean, while the standard deviation of the likelihood is fixed at the known data noise level of $0.1$. The noise level is intentionally kept small to ensure that any observed uncertainty arises primarily from the BNN.

**Inference details**  We use the Adam optimizer (Kingma & Ba, 2015) with a learning rate of $0.001$. We run SVGD, ASVGD, and SMI with five particles for 15,000 steps and OVI for 50,000 steps, sufficient for converging. We use 5,000 draws to estimate a performance metric for OVI and SMI. For both SMI and OVI, we use factorized Gaussians as guides. We use a hundred draws to estimate the Stein force for SMI (i.e., eq. (8)). All methods are initialized in $[-0.1, 0.1]$ and measurements are taken for ten different initialization. We use 500 warmup steps and a chain of 3000 steps for NUTS in NumPyro. All other parameters are default: max depth is 10, during warmup, the mass matrix is adapted using the online Welford scheme (Knuth, 1997) and step size calibrated using dual averaging (Nesterov, 2009).

### C.2.1 RECOVERY POINT SETUP

The recovery point experiment uses the same hyper-parameter setup as above. Recall that the recovery point is the number of particle SVGD required to get an LPPD below five particle SMI. To reach it, we begin with one particle SVGD and subsequently double until we reach the recovery point. We repeat the experiment ten times.

## C.3 MNIST CLASSIFICATION

This section outlines the details required to reproduce our MNIST classification results.

**Bayesian network**  We utilize both 2 and 3 hidden-layer BNNs of size 100 and tanh activation functions. Input images are flattened before being fed into the BNNs. The likelihood is modeled as a 10-class categorical distribution, parameterized by the logits produced by the BNN.

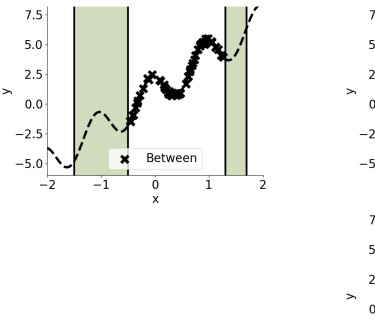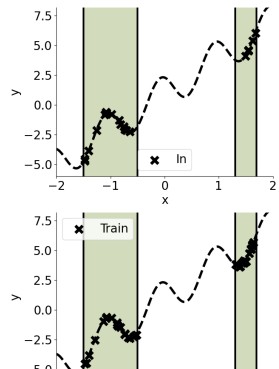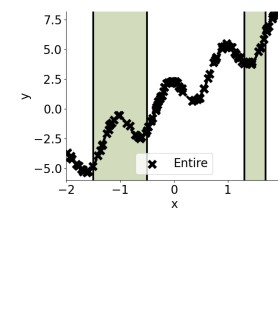

Figure 6: **Top row**: The samples were drawn from the data-generating process for evaluating Between, In and Entire regions. The In region used for inferring the BNNs is highlighted in grey. **Bottom row**: The samples drawn from the data-generating process to infer BNN posteriors.

**Inference details**  We employ the Adam optimizer with a learning rate of $10^{-3}$ for both MAP and OVI. For SVGD, SMI) and ASVGD, we use the Adagrad optimizer with a learning rate of $0.7$ for the 2 hidden-layer BNN and $0.8$ for the 3 hidden-layer BNN, utilizing five particles in each case. Specifically, the SMI method estimates the attractive force using $55$ draws.

All approaches are trained for $100$ epochs with a batch size of $128$. The images are scaled to [0,1]. Instead of random subsampling, we implement mini-batching and appropriately scale the likelihood, as described in Equation eq. (13).

## C.4 UCI REGRESSION BENCHMARK

In this section, we provide the details for reproducing our UCI regression results.

**Bayesian network**  We use a 2 hidden-layer Bayesian neural of size 50 and ReLU activation for all datasets. We use a Gaussian likelihood with the mean given by the BNN and a Gamma(shape=1, rate=0.1) prior on the precision (i.e., reciprocal variance). For SMI and OVI, we use the softplus $(x \mapsto \log(\exp(x) + 1))$ transformation on the Gaussian approximation to account for the difference in support of the likelihood precision. This is the transformation recommended in Kucukelbir et al. (2017) for inference using automatic differentiation when transforming a random variable from $R$ to $R^+$. The independent variable $(x)$ is standardized, while the dependent variable $(y)$ is kept as is. We randomly initialize guides uniformly in the interval $[-0.1, 0.1]$.

**Inferring the networks**  We randomly initialize the tested methods uniformly in unconstraint space within the interval $[-0.1, 0.1]$. This is lower than the NumPyro default of $[-2, 2]$. The initialization strategy mimics the initialization from Liu & Wang (2016) for SVGD and substantially reduces the steps needed for good performance.

We choose the learning rate from $[5 \cdot 10^{-i}]_{i=1}^{6}$ with a grid search on the first split of each data set. We select the learning rate with the best RMSE on a $10\%$ validation split from the training data. Table 6 provides the chosen learning rate for each method and dataset.

We use the Adam optimizer for up to 60.000 steps, inferring the BNNs a random subsample of size 100 without replacement. The independent variable $(x)$ is standardized, while the dependent variable $(y)$ is kept as is.

We use a convergence criterion on the Euclidean norm of $\phi^*$ for the particle methods. We compare a slow-moving norm average, calculated over the last 350 steps, against a fast-moving norm average, computed over the previous 35 steps. If the fast-moving average exceeds the slow-moving average, we conclude that the methods fluctuate around a minimum and stop iterating Equation (7). The number of past steps was chosen using the first split of the Boston Housing dataset with a learning rate of 0.5 using a 10% validation set from training to minimize RMSE.

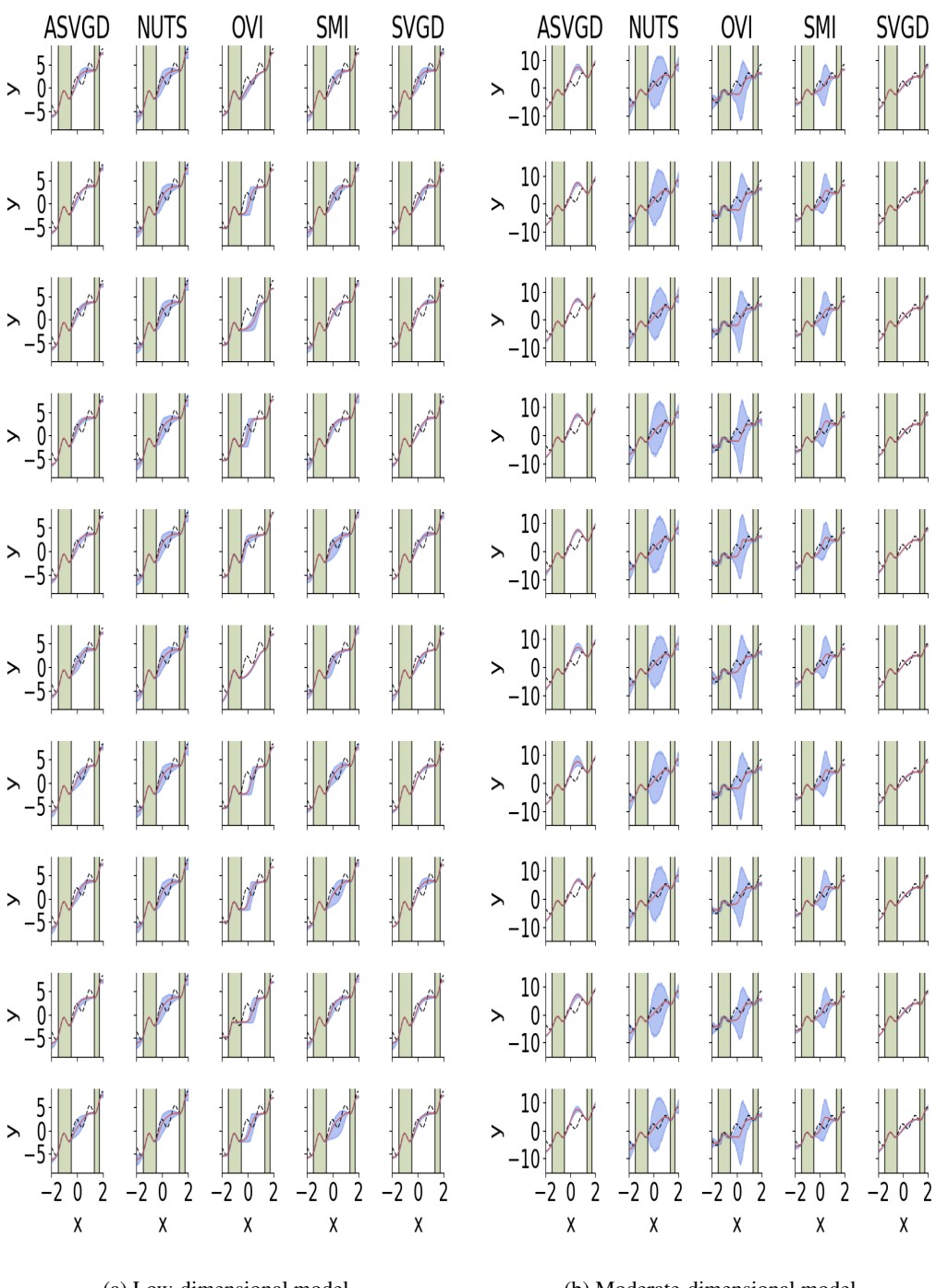

(a) Low-dimensional model          (b) Moderate-dimensional model

Figure 7: Figure 7a: High-density interval (HDI) for the low-dimensional model inferred using SMI, NUTS, SVGD, ASVGD and OVI on the 1D wave dataset (dotted line). SVGD, ASVGD, and SMI use five particles. The posteriors are inferred with data drawn from the In region, highlighted with vertical lines. Figure 7b: HDI for the moderate-dimensional model. ASVGD and SVGD display collapse by a significant narrowing in HDI between the In regions when comparing the low to moderate dimensions. In low-dimensional models, initialization plays a role in narrowing or widening HDI for all methods. In mid-sized models, SMI is robust to initialization.

Table 6: Learning rate for the methods used on the standard and Gap10 splits of UCI.

| Dataset | ASVGD | MAP | OVI | SMI | SVGD |
|---------|-------|-----|-----|-----|------|
| Standard UCI | | | | | |
| Boston | $5 \cdot 10^{-5}$ | $5 \cdot 10^{-5}$ | $5 \cdot 10^{-5}$ | $5 \cdot 10^{-5}$ | $5 \cdot 10^{-6}$ |
| Concrete | $5 \cdot 10^{-4}$ | $5 \cdot 10^{-4}$ | $5 \cdot 10^{-3}$ | $5 \cdot 10^{-4}$ | $5 \cdot 10^{-2}$ |
| Energy | $5 \cdot 10^{-4}$ | $5 \cdot 10^{-4}$ | $5 \cdot 10^{-3}$ | $5 \cdot 10^{-4}$ | $5 \cdot 10^{-4}$ |
| Kin8nm | $5 \cdot 10^{-5}$ | $5 \cdot 10^{-5}$ | $5 \cdot 10^{-4}$ | $5 \cdot 10^{-3}$ | $5 \cdot 10^{-4}$ |
| Naval | $5 \cdot 10^{-5}$ | $5 \cdot 10^{-4}$ | $5 \cdot 10^{-4}$ | $5 \cdot 10^{-4}$ | $5 \cdot 10^{-5}$ |
| Power | $5 \cdot 10^{-4}$ | $5 \cdot 10^{-4}$ | $5 \cdot 10^{-3}$ | $5 \cdot 10^{-4}$ | $5 \cdot 10^{-4}$ |
| Protein | $5 \cdot 10^{-5}$ | $5 \cdot 10^{-3}$ | $5 \cdot 10^{-3}$ | $5 \cdot 10^{-3}$ | $5 \cdot 10^{-3}$ |
| Wine | $5 \cdot 10^{-5}$ | $5 \cdot 10^{-5}$ | $5 \cdot 10^{-3}$ | $5 \cdot 10^{-3}$ | $5 \cdot 10^{-5}$ |
| Yacht | $5 \cdot 10^{-5}$ | $5 \cdot 10^{-5}$ | $5 \cdot 10^{-3}$ | $5 \cdot 10^{-3}$ | $5 \cdot 10^{-5}$ |
| Gap10 UCI | | | | | |
| Boston | $5 \cdot 10^{-5}$ | $5 \cdot 10^{-5}$ | $5 \cdot 10^{-5}$ | $5 \cdot 10^{-3}$ | $5 \cdot 10^{-2}$ |
| Concrete | $5 \cdot 10^{-4}$ | $5 \cdot 10^{-4}$ | $5 \cdot 10^{-3}$ | $5 \cdot 10^{-4}$ | $5 \cdot 10^{-3}$ |
| Energy | $5 \cdot 10^{-4}$ | $5 \cdot 10^{-2}$ | $5 \cdot 10^{-2}$ | $5 \cdot 10^{-4}$ | $5 \cdot 10^{-4}$ |
| Kin8nm | $5 \cdot 10^{-5}$ | $5 \cdot 10^{-4}$ | $5 \cdot 10^{-3}$ | $5 \cdot 10^{-5}$ | $5 \cdot 10^{-4}$ |
| Naval | $5 \cdot 10^{-5}$ | $5 \cdot 10^{-4}$ | $5 \cdot 10^{-4}$ | $5 \cdot 10^{-4}$ | $5 \cdot 10^{-5}$ |
| Power | $5 \cdot 10^{-4}$ | $5 \cdot 10^{-5}$ | $5 \cdot 10^{-2}$ | $5 \cdot 10^{-4}$ | $5 \cdot 10^{-4}$ |
| Protein | $5 \cdot 10^{-4}$ | $5 \cdot 10^{-3}$ | $5 \cdot 10^{-3}$ | $5 \cdot 10^{-4}$ | $5 \cdot 10^{-3}$ |
| Wine | $5 \cdot 10^{-5}$ | $5 \cdot 10^{-5}$ | $5 \cdot 10^{-4}$ | $5 \cdot 10^{-2}$ | $5 \cdot 10^{-5}$ |
| Yacht | $5 \cdot 10^{-5}$ | $5 \cdot 10^{-3}$ | $5 \cdot 10^{-3}$ | $5 \cdot 10^{-3}$ | $5 \cdot 10^{-4}$ |

Table 7: Summary statistics for the standard UCI benchmark datasets with train-test splits from Hernández-Lobato & Adams (2015) and Gap10 benchmark datasets adapted from Foong et al. (2019) to use $10\%$ for testing instead of $33\%$.

| Dataset | Train size | Test size | Features | Std Splits | Gap10 Splits |
|---------|-----------|-----------|----------|-----------|--------------|
| Boston | 455 | 51 | 13 | 20 | 13 |
| Concrete | 927 | 103 | 8 | 20 | 8 |
| Energy | 691 | 77 | 8 | 20 | 8 |
| Kin8nm | 7373 | 819 | 8 | 20 | 8 |
| Naval | 10741 | 1193 | 17 | 20 | 17 |
| Power | 8611 | 957 | 4 | 20 | 4 |
| Protein | 41157 | 4573 | 9 | 5 | 9 |
| Wine | 1439 | 160 | 11 | 20 | 11 |
| Yacht | 277 | 31 | 6 | 20 | 6 |

**The standard UCI split** We use the train-test splits from Mukhoti et al. (2018) for our standard UCI results. Table 7 gives summary statistics of the datasets. We treat features and responses (i.e., $(x, y)$) as real values.

**The Gap10 UCI split** We use the methodology suggested in Foong et al. (2019) to construct the GAP dataset. We sort each feature dimension individually, taking the middle tenth as a test and leaving the two tails as our training split. Using this procedure will result in as many splits as features. However, where Foong et al. (2019) allocated the middle third for testing, we use a tenth to have the same test allocation as standard UCI. Comparing Standard to Gap10 in table 7, the Gap10 generally produces fewer splits than standard UCI.

**Time comparison for VI methods** In table 8, we reproduce the per-step average inference time [sec/step] on the UCI datasets for SMI, SVGD, ASVGD, OVI and MAP. On UCI datasets, SMI exhibits slower inference compared to the VI-based baselines. A portion of this overhead arises from JIT compilation, which we believe can be reduced by optimizations in future releases of SMI.

Table 8: The table shows the average time per step (in seconds per step) for datasets in the UCI regression benchmark. Although SMI demonstrates slower inference times than alternative methods, the recovery point experiment indicates that SMI offers a more favorable trade-off when considering the associated performance improvements.

| Dataset | SMI | SVGD | ASVGD | OVI | MAP |
|---------|------|------|-------|------|------|
| Boston | 0.0014 | 0.0003 | 0.0003 | 0.0002 | 0.0001 |
| Concrete | 0.0015 | 0.0004 | 0.0003 | 0.0002 | 0.0001 |
| Energy | 0.0017 | 0.0003 | 0.0003 | 0.0002 | 0.0001 |
| Kin8nm | 0.0192 | 0.0004 | 0.0004 | 0.0003 | 0.0002 |
| Naval | 0.0103 | 0.0004 | 0.0004 | 0.0004 | 0.0001 |
| Power | 0.0079 | 0.0004 | 0.0004 | 0.0002 | 0.0002 |
| Protein | 0.0468 | 0.0011 | 0.0008 | 0.0004 | 0.0003 |
| Wine | 0.0093 | 0.0003 | 0.0003 | 0.0002 | 0.0001 |
| Yacht | 0.0059 | 0.0003 | 0.0003 | 0.0003 | 0.0001 |

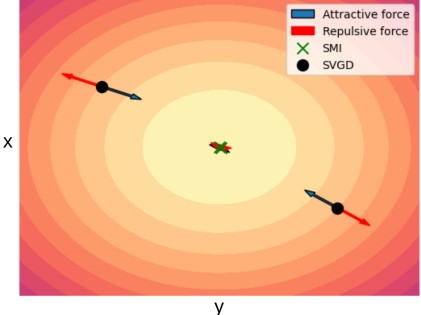

(a) Location dimension of particles.

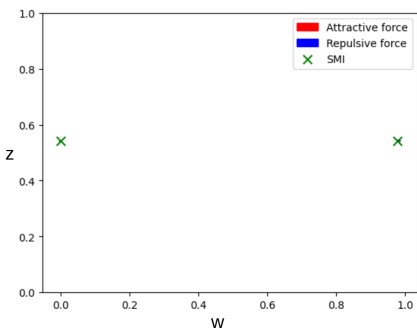

(b) Variance dimension of particles.

Figure 8: Converged two-particle approximation of a two-dimensional Gaussian using SMI and SVGD. Each SMI particle, denoted as $\boldsymbol{\psi}$, parameterizes a Gaussian guide with $\boldsymbol{\psi} = (x, y, z, w)$, where $(x, y)$ represent the guide's location and $(z, w)$ represent its variances. In contrast, an SVGD particle, denoted as $\boldsymbol{\theta}$, only represents location dimensions, i.e., $\boldsymbol{\theta} = (x, y)$. **Left**: Location dimensions of SMI and SVGD particles. Shades show the equiprobability contours of the Gaussian. **Right**: Variance dimensions of the SMI particles. SVGD particles are absent here as they only represent location. SMI effectively approximates the Gaussian by explicitly incorporating variance as part of its particle dimensions. The SMI forces are scaled for better visibility in the location dimensions. No force arrows are visible in the variance dimensions because the system has converged, making the forces negligible.

When considering the recovery point experiment table 4, SMI demonstrates significantly improved runtime efficiency. On the mid-sized network, SMI achieves inference times 6x faster than SVGD. This observation suggests that while VI methods excel in runtime on UCI datasets, SMI provides a better trade-off when factoring in performance gains. Thus, in contexts where accuracy and robustness are critical, SMI is preferable despite its higher initial runtime cost.

# D SMI VERSUS SVGD: INSIGHTS FROM A SIMPLE TOY MODEL

To address variance collapse, the key distinction between SVGD and SMI lies in the space the particles occupy. SMI particles operate in a higher-dimensional space than SVGD particles. This allows the repulsive term, $\nabla_1 k(x, y)$, in SMI to influence the distribution's shape and its parameterized location. In contrast, SVGD particles can only control location.

In SVGD, each particle represents a latent parameter sample. Meanwhile, in SMI, each particle parameterizes an entire distribution. For example, if the parameterized distribution is a factorized Gaussian, each SMI particle would represent both the mean (location) and variance of the Gaussian.

While the location component of an SMI particle shares the same space as an SVGD particle, the variance component has no equivalent in SVGD. As a result, the repulsive force in SMI operates in a broader space, encompassing both location and variance.

This distinction becomes evident when comparing SVGD and SMI in a two-particle approximation of a standard Gaussian distribution. The SMI approximation forms a Gaussian mixture. By breaking SMI particles into their location and variance components, we can visualize the location component within the same space as SVGD particles and the target Gaussian density. In fig. 8a, SMI particle locations converge toward the center of the target Gaussian, while SVGD particles spread out, maintaining equal distances from the Gaussian center.

At first glance, focusing solely on the location dimension of the SMI particles might suggest they have collapsed. However, this interpretation is incomplete because it overlooks the role of variance. In fig. 8b, we examine the variance component of the SMI particles. We see that the two-particle SMI system together captures the variance of the target density because each dimension in fig. 8b sums to one. Consequently, the SMI estimation has not collapsed. Instead, it captures the target with all four dimensions of its particles (two for location and two for variance) rather than the two dimensions of the SVGD particle.

