# OpenReview forum: "ELBOing Stein: Variational Bayes with Stein Mixture Inference"
_ICLR.cc/2025/Conference — ICLR 2025 Poster_

### Official Review · Reviewer_dciU · 2024-10-27

**Soundness:** 3
**Presentation:** 3
**Contribution:** 3
**Rating:** 6
**Confidence:** 3

**Summary:**

The posterior of a Bayesian model is sometimes intractable, calling for approximate inference techniques. This paper focuses on the idea of approximating the Bayesian posterior with a mixture distribution where each component is parameterized separately but still in the same family. By viewing the ELBO as an objective with permutation invariant parameters, this paper incorporates ideas from Nonlinear-SVGD (NSVGD) and develop a Stein-style update rule of the mixture parameters. The resulted method, called Stein mixture inference (SMI), prevents itself from variance collapse. The paper also shows that asymptotically the optimized bound is an ELBO.

In the experiments, it is first shown that Stein-based methods suffer from variance collapse in synthesized Gaussian models and 1D regression models. In contrast, SMI and vanilla VI (OVI) produce the desired uncertainty. On the UCI regression benchmarks, SMI gives the best NLL in the most cases, comparing with other Stein-based methods and OVI.

**Strengths:**

I believe the method is novel and the main idea is sound. The claims are clearly presented and supported by empirical evidences. Overall it is a complete work with a few concerns.

**Weaknesses:**

This paper has the title of ELBOing Stein, but I would rather call it Steining ELBO, which seems a bit unnecessary. To convince me to increase my score, I would like to see a discussion of [1], which also uses a mixture distribution to approximate the posterior.

If one is using a mixture distribution to target the Bayesian posterior, the most direct approach would be to try VI, instead of deriving a complicated Stein-based method. One pro of VI is that the mixture weights can be adjusted while one con is that it does not fully make use of the exchangeability of parameters. However, this work only considers mean-field VI, which is a really weak baseline. I would like to see how the permutation invariance helps the optimization of the mixture distribution.

It is not surprising that optimizing an VI objective prevents the approximate posterior from the pitfalls of SVGD. As shown in the paper, OVI does not have the issue. The argument that "SMI is more particle-efficient than SVGD" is translated to me as "VI with mixtures is more particle-efficient than SVGD". Then what is the point of using Stein?

Line 150 says that "Particle methods are attractive due to their freedom from strong parametric assumptions". Mixture distribution in this paper seems to be a strong parametric assumption, especially when it uses fewer particles than SVGD, which further drags this work away from Stein.

The experiment section is also rather weak. The benchmark models all have very low dimensions. I expect a Bayesian inference algorithm in 2024 to be tested on more recent benchmarks, like models in posteriorDB, or larger BNN problems.

[1] Morningstar, W., Vikram, S., Ham, C., Gallagher, A., & Dillon, J. (2021, March). Automatic differentiation variational inference with mixtures. In International Conference on Artificial Intelligence and Statistics (pp. 3250-3258). PMLR.

**Questions:**

- How does SMI compare with ADVI with mixtures?
- How is each component in the mixture distributions parameterized?

---

> ### Author Response · Authors · 2024-11-16
> **Rebuttal**
>
> ## To convince me to increase my score, I would like to see a discussion of [1], which also uses a mixture distribution to approximate the posterior.
> Thank you for highlighting this paper. We will add a discussion of ADVI with mixtures to the section Related Work in the article.
>
> ## How does SMI compare with ADVI with mixtures?
> In the article you highlighted, the authors propose two objective functions: SIELBO and SIWAE. SIELBO is similar to only using the "attractive force" of SMI (i.e., let the mixture weight alpha_k in [1] be the kernel evaluation and use the ELBO from line 273), which they point out suffers from particle collapse. They address this issue by introducing importance weighting, which leads to the SIWAE objective
>
> In contrast, we tackle particle collapse by introducing a regularizer on the particles using the reproducing kernel. Our method can easily incorporate the SIWAE objective by adding importance weights to the ELBO of the attractive force. Since they have already demonstrated that this is an ELBO, it will be straightforward to show that, with our entropic regularizer, we still maintain the ELBO property.
>
> The key difference between SIWAE and our method is that SIWAE's advancements conclude at this point. However, with SMI, we have a vast design space of kernels to explore, which may allow us to scale this method to larger models. The current article provides the necessary foundation for this further body of work (see below).
>
> ## The experiment section is also rather weak. The benchmark models all have very low dimensions. I expect a Bayesian inference algorithm in 2024 to be tested on more recent benchmarks, like models in posteriorDB, or larger BNN problems.
>
> Our experiments are designed to support our claim that SMI can mitigate variance collapse in SVGD. To this end, we are already considering models substantially larger than those used in previous work [2,3]  on variance collapse in SVGD. By demonstrating SMI's robustness in moderately-sized models, we establish a solid foundation and necessary precondition for its extension to high-dimensional, real-world applications, which will require additional work on issues such as hyperparameter choice (notably regarding the choice of the kernel, which potentially provides a way to tackle the important open issue of the non-identifiability of deep models).
>
> ## How is each component in the mixture distributions parameterized?
> In a Gaussian mixture model (GMM), the overall distribution is represented as a combination of multiple Gaussian components, each of which is defined by specific parameters—namely, a mean and a variance. In SMI, each particle refers to a vector containing the mean and variance of one Gaussian component, effectively parameterizing it. Thus, a particle in this context is the pair (mean, variance) that defines the central location and spread of that Gaussian. When multiple particles are used, each representing a different Gaussian with unique parameters, they collectively form the GMM, with each particle contributing one component to the mixture. This setup allows the mixture model to flexibly approximate complex distributions by combining the influences of multiple Gaussian distributions, each defined by a particle.
>
> ## What is the point of using Stein?
>   1. Optimizing a variational distribution that is a mixture model is inherently challenging, and NSVGD provides a principled and tractable framework for this task.
>   2. By leveraging NSVGD, SMI connects to a rich field of theoretical results, such as its relationship with Wasserstein flows, which provide a sound methodological foundation and allow for future research and innovation.
>   3. Mixture models are universal approximators of smooth densities. Consequently, SMI retains considerable freedom from strong parametric assumptions compared to methods relying on a single variational distribution, thereby preserving the flexibility of Stein-based methods.
>   4. The kernel component in SMI is a versatile tool that allows users to incorporate inductive biases into the variational approximation. For instance, the non-identifiability of deep models is a significant unresolved challenge in their Bayesian estimation [4]. By choosing suitable kernels, SMI offers the potential to mitigate such issues.
>
> ## References
>
>   1. Morningstar, W., Vikram, S., Ham, C., Gallagher, A., & Dillon, J. (2021, March). Automatic differentiation variational inference with mixtures. In International Conference on Artificial Intelligence and Statistics (pp. 3250-3258). PMLR.
>   2. Ba, Jimmy, et al. "Understanding the variance collapse of SVGD in high dimensions." International Conference on Learning Representations. 2021.
>   3. Zhuo, Jingwei, et al. "Message passing Stein variational gradient descent." International Conference on Machine Learning. PMLR, 2018.
>   4. Roy, Hrittik, et al. "Reparameterization invariance in approximate Bayesian inference." NeurIPS. 2024.

---

> > ### Comment · Reviewer_dciU · 2024-11-20
> >
> > Thank you for the rebuttal and the explanation of the entropy term (the repulsive force). I believe the discussion of mixture-based VI will demonstrate the position of this work and strengthen the contributions. I do not have additional questions and am glad to increase my score.

---

### Official Review · Reviewer_HpKW · 2024-11-02

**Soundness:** 2
**Presentation:** 2
**Contribution:** 3
**Rating:** 6
**Confidence:** 3

**Summary:**

- The authors propose a method called Stein Mixture Inference (SMI) to address the issue of variance collapse observed in Stein Variational Gradient Descent (SVGD).
- SMI extends the Nonlinear SVGD framework (Wang & Liu, 2019) to variational inference (VI) by allowing each particle to parameterize a component distribution within a mixture model, thereby introducing ELBO-like objectives.
- The authors show that SMI offers several advantages over standard SVGD by effectively mitigating variance collapse.

**Strengths:**

- The application of variational inference (VI) concepts to Stein Variational Gradient Descent (SVGD) appears novel and intriguing.
- The authors validate their VI-based approach through numerical experiments on several UCI benchmark datasets, demonstrating good performance. The results seem to suggest that this approach effectively mitigates the impact of variance collapse.

**Weaknesses:**

### Insufficient Analysis of the Motivation Behind Extending SVGD with VI for Variance Collapse Mitigation:
- The main objective of this paper, as I understand it, is to mitigate variance collapse by extending the SVGD objective function through a combination of an ELBO-like objective from VI and the Non-linear SVGD framework. However, it is not entirely clear "why" this extension effectively mitigates variance collapse. While Figure 1 provides a conceptual illustration, it does not intuitively explain how the proposed method addresses variance collapse. Additionally, while the third paragraph of the Introduction discusses the motivation for this approach, it remains unclear how using a mixture of approximate distributions around SVGD particles with a VI-inspired objective avoids variance collapse.
- The authors propose controlling particles through variational distributions, similar to VI, as a solution to the variance collapse issue. However, given the use of the NSVGD framework, the critical role of this aspect remains unclear. The entropy regularization term could potentially affect not only mode collapse but also variance collapse. If the VI-inspired approach is indeed effective, the method should perform well even with $\alpha=0$. In this context, Figure 2 shows that variance collapse is mitigated even when $\alpha$ takes small values, suggesting that particle control via variational distributions may be effective. On the other hand, this result implies that regularization may not play a significant role, raising questions about the necessity of combining it with NSVGD. Overall, it remains unclear why the NSVGD framework is essential and which part of the proposed approach effectively addresses variance collapse.

### Concerns Regarding the Limited Number of Comparative Methods:
- For sample approximations of the posterior distribution, methods such as HMC (Neal, 2011) and MMD descent (Arbel et al., 2019; Ba et al., 2021) are also effective. However, this study only compares performance within the SVGD family of methods and EVI, leaving questions about the extent to which the proposed method mitigates variance collapse in the broader context of approximate inference. Given that (Ba et al., 2021) also includes these methods in numerical experiments addressing variance collapse, this comparison is essential for validating contributions in this research area.
- Additionally, the absence of a comparison with the resampling method proposed by (Ba et al., 2021) raises concerns regarding the integrity of the performance evaluation. While the authors argue in Section 5 that the resampling method is computationally infeasible, I believe this does not fully justify its exclusion as a comparative method. Given the availability of an “NVIDIA Quadro RTX 6000 GPU,” running such methods may not be computationally prohibitive, at least for datasets like the UCI benchmarks.
- Furthermore, I find it difficult to agree with the authors’ claim: “Annealing SVGD (ASVGD) D’Angelo & Fortuin (2021a) is the only alternative that directly addresses variance collapse in SVGD with a viable method.” I believe that the resampling method proposed by (Ba et al., 2021) is also aimed at mitigating the variance collapse problem.

### Citation:
- (Neal, 2011): R. M. Neal. MCMC using Hamiltonian dynamics. Handbook of Markov Chain Monte Carlo, 2(11):2. https://arxiv.org/abs/1206.1901.
- (Arbel et al., 2019): M. Arbel, A. Korba, A. Salim, and Arthur Gretton. Maximum Mean Discrepancy Gradient Flow. NeurIPS2019. https://arxiv.org/abs/1906.04370.

**Questions:**

- Could you provide additional analysis or intuition on why the combination of an ELBO-like objective from VI and the Non-linear SVGD framework effectively mitigates variance collapse? Specifically, how does modeling a mixture of approximate distributions around the vicinity of SVGD particles help in avoiding variance collapse? A more detailed explanation or visual representation would be appreciated.
  - For example, could you provide a more detailed explanation or visual representation of how the mixture components interact with the ELBO objective to mitigate variance collapse? For instance, a step-by-step explanation or diagram illustrating how the proposed method addresses the variance collapse problem would be helpful.
- Why is the integration with the NSVGD framework necessary in your method? Is there evidence that the entropy regularization term alone is insufficient to address variance collapse? Given that Figure 2 shows variance collapse is mitigated even when $\alpha$ takes small values, does this imply that the regularization component may not be as critical? If so, what is the rationale for including it in the framework?
- Why were methods such as HMC and MMD descent not included in the comparative analysis, especially given their relevance in approximate inference and their use in experiments in (Ba et al., 2021)?
  - If possible, could you add comparisons with HMC (Neal, 2011) and MMD descent (Arbel et al., 2019; Ba et al., 2021) in the experimental section, particularly at least on the UCI datasets, to provide a broader context for evaluating SMI’s performance in addressing variance collapse? If a full comparison is not feasible, could you discuss how SMI might be expected to compare to these methods theoretically or empirically, based on existing literature?
- Could you elaborate on why the resampling method from (Ba et al., 2021) was excluded as a comparative method, despite the computational resources available (e.g., “NVIDIA Quadro RTX 6000 GPU”)? Is this method genuinely computationally infeasible for UCI benchmark datasets, or were there other factors influencing its exclusion?
  - So, could you include the resampling method from Ba et al. (2021) in your comparisons, particularly on the UCI datasets, to strengthen the evaluation? If this is not feasible, could you provide a more detailed justification for why it is computationally infeasible, even with the available GPU resources? Additionally, if an empirical comparison is truly not possible, could you discuss how SMI theoretically compares to the resampling approach?

**Details Of Ethics Concerns:**

N/A.

---

> ### Author Response · Authors · 2024-11-16
> **Rebuttal**
>
> ## Could you provide additional analysis or intuition on why the combination of an…
>
> To address variance collapse, the key distinction between SVGD and SMI lies in the space the particles occupy. SMI particles operate in a higher-dimensional space than SVGD particles. This allows the repulsive term ($\nabla_1 k(x,y)$) in SMI to influence both the shape of the distribution and its parameterized location. In contrast, SVGD particles can only control location.
>
> In SVGD, each particle represents a latent parameter sample. Meanwhile, in SMI, each particle parameterizes an entire distribution. For example, if the parameterized distribution is a factorized Gaussian, each SMI particle would represent both the mean (location) and variance of the Gaussian. While the location component of an SMI particle shares the same space as an SVGD particle, the variance component has no equivalent in SVGD. As a result, the repulsive force in SMI operates in a broader space, encompassing both location and variance.
>
> This distinction becomes evident when comparing SVGD and SMI in a two-particle approximation of a standard Gaussian distribution. The SMI approximation forms a Gaussian mixture. By breaking SMI particles into their location and variance components, we can visualize the location component within the same space as SVGD particles and the target Gaussian density. In this visualization, SMI particle locations converge toward the center of the target Gaussian, while SVGD particles spread out, maintaining equal distances from the Gaussian center.
>
> [Mean estimation image](https://storage.googleapis.com/iclr25_suppl/means.png).
>
> At first glance, it might seem that SMI particles have collapsed when considering only their locations. However, this interpretation is incomplete because it only tells half the story. When we examine the variance component of the SMI particles, we observe that a single SMI particle captures the variance in one dimension of the target Gaussian distribution, and both particles cover the variance in the other dimension. **We will add this example to the appendix.**
>
> [Variance estimation image](https://storage.googleapis.com/iclr25_suppl/variances.png).
>
> Next, we explore the challenges posed by the RBF kernel in SMI, which exhibits certain pathological behaviors. To extend SMI to larger models, we must carefully design new kernels. Ideally, these kernels should account for the volume of the distribution parameterized by each SMI particle. We believe that probability product kernels, along with concepts from linear-Laplace approximations, will enable us to address these issues effectively.
>
> Scaling SMI to large models, however, is a substantial research endeavor that requires dedicated study and goes beyond the scope of this work. This paper serves as foundational work, demonstrating that SMI can resolve variance collapse issues inherent to SVGD. The contributions here should be evaluated on this primary achievement.
>
> ## Is there evidence that the entropy regularization term alone is insufficient to address variance collapse?
> Yes. This corresponds to having the score function in the attractive force. With this substitution, we are doing SVGD (as shown in Appendix A3.1), which suffers from variance collapse, as demonstrated in the estimating Gaussian variance experiment and 1D regression with synthetic data.
>
> ##  SVGD with resampling (Ba et al. 2021)
> We omitted SVGD with resampling (reSVGD), Algorithm 1, of Ba et al. (2021) in our Gaussian variance estimation experiment because in reproducing their results on the Gaussian variance estimation, we found that their proposed algorithm gives a biased estimator (i.e., $E[X \sim \text{reSVGD}] \not = 0]$). This is not the case for SMI, SVGD or ASVGD. **We will include the reproduced result of their method for the Gaussian variance estimation and add the plot showing that reSVGD is biased in the appendix.**
>
> For the 1D and UCI experiments, we did not include a comparison with the Ba et al. (2021) algorithm due to several limitations. First, there are no publicly available implementations of their method, and the paper explicitly states that it is an analytic tool, not scalable due to its $O(m^4)$ computational complexity.
>
> Second, the algorithm is inherently sequential, making it poorly suited for GPU acceleration, as it limits parallelization. Additionally, the memory overhead is substantial—each step requires sampling and storing a new set of particles, which can quickly lead to out-of-memory errors. Furthermore, frequent memory access on GPUs significantly diminishes performance gains.
>
> Finally, their algorithm relies on knowing the variance to maintain during resampling, which is straightforward when the target is a standard Gaussian but impractical for the posterior of a Bayesian Neural Network (BNN). We did not include reSVGD in the UCI and 1D regression experiments for these reasons.

---

> ### Author Response · Authors · 2024-11-16
> **Rebuttal**
>
> We would like to clarify that Figure 2 does **not** imply that variance **collapse** is mitigated with a small alpha ($0<\alpha\ll1$) for SMI in the general case.
>
> In Figure 2 (**right**), we are using one particle, so the choice of alpha does not affect this experiment. We show this theoretically and experimentally. The theoretical validation is in Appendix A.3.2, where we show that one particle SMI is MFVI, so the objective SMI is the regular ELBO from eq. 4 (which contains no alpha). It is empirically validated in Figure 2 **right**, where the variance estimation is one regardless of $\alpha$.
>
> In Figure 2 (**left** and **middle**), we investigate an artificial pathological case by using an overly large number of particles (20). That is, the variational model is much richer than warranted by the data. In this case, SMI will actually **overestimate** variance. However, we can still get good results for SMI by setting alpha to a small value (which mitigates the **overestimation** of variance in this pathological case). In the general case, this does not apply, and the entropy terms remain important.
>
> ## HMC Baseline
> Adding HMC as a baseline is an excellent suggestion. **We will provide results on HMC with NUTS for the UCI datasets and 1D regression tasks.** As expected, NUTS is comparatively slow but its prediction performance is good. We will include NUTS as a gold standard for comparison in the article and provide the results as an official comment.

---

> > ### Comment · Reviewer_HpKW · 2024-11-22
> > **Reply for Rebuttal**
> >
> > Thank you very much for your thoughtful response.
> > - Regarding the additional explanation about the combination of an ELBO-like objective from VI and Non-linear SVGD, I feel that you have provided a convincing discussion. I did not intend to suggest that the proposed methodology in this study should be extended to large-scale models, and I agree that this could be one of the future directions of research.
> > - I have one question regarding the images you provided. Could you kindly explain how these should be interpreted? Specifically, I found the second image (the one illustrating variance estimation) a bit difficult to understand. If the figure is animated, I was unfortunately unable to view it on my end.
> > - Thank you for pointing me to the appendix for evidence that the entropy regularization term alone is insufficient. Personally, I believe that including this discussion in the main body of the paper would help readers better understand the content of the paper (although I leave it to your discretion whether to revise this).
> > - If evidence that reSVGD is biased is added as a figure in the appendix, I understand why Ba et al. (2021) was excluded from the experiments. Additionally, I found your explanation that the method is unsuitable for GPUs highly informative, as it was something I had not considered. Thank you for addressing my misunderstanding of the experimental results so clearly. Including an explanation similar to this in the paper would improve its readability.
> > - I am delighted to hear that an experiment using HMC as a baseline will be added. I believe this will further clarify the positioning of the proposed method.
> >
> > Given the above considerations, most of my concerns have been appropriately addressed. Therefore, I would like to update my score to 6.

---

> > > ### Author Response · Authors · 2024-11-25
> > > **Reply**
> > >
> > > ## Could you kindly explain how these should be interpreted?
> > >
> > > The key insight from the second image is that SMI's repulsion mechanism includes additional dimensions that are not present in SVGD. In particular, when using Gaussian component distributions in SMI, the variance of each component acts as an extra dimension, which has no counterpart in SVGD. The plot shows how the variance of each particle is represented when SMI approximates a 2-dimensional target Gaussian distribution. The force arrows in the plot are tiny and barely visible because the system has already properly converged, consequently resulting in minimal forces acting on the particles.

---

> > > > ### Comment · Reviewer_HpKW · 2024-11-30
> > > > **Acknowledgement**
> > > >
> > > > Thank you for your detailed explanation.
> > > > Now my concerns have been addressed.
> > > >
> > > > I find this paper interesting.
> > > > Best wishes for success of your paper.
> > > >
> > > > Sincerely,
> > > > --Reviewer HpKW

---

### Official Review · Reviewer_mFTt · 2024-11-03

**Soundness:** 3
**Presentation:** 3
**Contribution:** 3
**Rating:** 6
**Confidence:** 2

**Summary:**

The paper introduces Stein Mixture Inference (SMI), which optimizes a lower bound to the Evidence Lower Bound (ELBO).
SMI extends Nonlinear Stein Variational Gradient Descent (NSVGD) to the variational Bayes setting and addresses the issue of variance collapse.
The effectiveness of SMI is demonstrated on both synthetic and real-world datasets.

**Strengths:**

1. The problem addressed in this paper is both important and compelling. Traditional approaches like ordinary mean-field variational inference (OVI) and Stein Variational Gradient Descent (SVGD) often experience variance collapse, whereas SMI provides more accurate variance estimates, improving uncertainty quantification.

2. The paper is well-written, providing a clear background and a thorough summary of related work. As someone slightly unfamiliar with the field, I particularly appreciated the authors' effort to re-explain and contextualize prior results, which greatly helped in assessing the paper's contributions.

3. SMI is compared with other methods across a variety of synthetic and real-world datasets.

**Weaknesses:**

1. Variational inference offers a compelling alternative to sampling methods like MCMC due to its efficiency, especially in high-dimensional settings and with large-scale datasets.
However, the current validation of SMI is limited to small to moderately-sized models, which somewhat limits its appeal and persuasiveness for broader, large-scale applications.

2. The paper lacks theoretical insights or guidance on how SMI’s performance depends on the number of particles $m$.
Providing recommendations or analysis on selecting an appropriate particle count $m$ would greatly enhance its practical applicability.

**Questions:**

1. It’s challenging to distinguish the lines representing different methods in Figure 2 (e.g. $SMI_{1}$, $SMI_{20}$).
Using distinct colors for each method would improve the visualization and make the differences clearer.
2. The experiments in Section 6.1 demonstrate that SMI overcomes variance collapse. It would also be valuable to assess whether the approximate distribution given by SMI accurately captures the shape of the posterior.
This could be evaluated by comparing the estimated covariance matrix with the target covariance matrix.

---

> ### Author Response · Authors · 2024-11-19
> **Rebuttal**
>
> ## W1: Variational inference offers a compelling alternative to sampling methods like MCMC due to its efficiency...
>
> We appreciate the reviewer’s observation regarding the scope of our experiments. Our primary objective in this work is to introduce SMI and show that it effectively addresses the issue of variance collapse in SVGD. To this end, we focus on models that are already significantly larger than those typically considered in prior studies on variance collapse in SVGD (and that appeared in ICML and ICLR [1,2]). Our current contribution thus introduces a new method and demonstrates its robustness in addressing this specific challenge.
>
> Scaling SMI to large-scale models is indeed important but extends beyond the scope of this paper. However, we argue that it is clear that SMI provides ample opportunities for scaling to larger models, notably through the use of bespoke kernels that go beyond simple RBF kernels, building inductive bias into the guide, hyperparameter tuning and so on. We plan to explore these opportunities in a future publication.
>
> ## W2: The paper lacks theoretical insights or guidance on how SMI’s performance depends on the number of particles...
>
> We do suggest using alpha to check if the approximation is too rich. This heuristic consists in checking if the variance (or calibration) of the estimator is invariant for values of alpha << 1. If the variance is stable, it suggests that too many particles are used, allowing inference with fewer particles without compromising uncertainty estimation. Beyond this heuristic, standard hyperparameter optimization techniques, such as cross-validation, dataset splitting, or leveraging information criteria, can also guide the selection of particle count.
>
> We acknowledge, however, that developing a rigorous theoretical framework to determine the optimal number of particles remains an open and intriguing avenue for future research. Such work could include deriving bounds on variance or error as functions of particle count and alpha, which would offer deeper insights into this critical aspect of SMI. In this respect, the considerable body of theoretical results concerning NSVD and Stein’s method provides a solid ground. However, this is beyond the scope of this article, which introduces SMI and demonstrates that SMI can be used to mitigate variance collapse in SVGD.
>
> ## Q1: It’s challenging to distinguish the lines representing different methods in Figure 2 (e.g. , )...
>
> Thank you for the recommendation. We will update Figure 2 with distinct colors.
>
> ## Q2: The experiments in Section 6.1 demonstrate that SMI overcomes variance collapse...
>
> Our target posterior is a multivariate standard Gaussian, and the SMI approximation is modeled as a Gaussian Mixture Model (GMM) with mean-field Gaussian components. This setup is designed to align the posterior approximation with the true posterior. To further demonstrate this, **we will include a plot in the appendix comparing the Frobenius norm between the estimated covariance matrices and the target covariance matrix.** This will provide a quantitative measure of how well the SMI approximation captures the shape of the posterior.
>
> [Posterior shape plot.](https://storage.googleapis.com/iclr25_suppl/post_shape.png)
>
> ## References
> 1. Ba, Jimmy, et al. "Understanding the variance collapse of SVGD in high dimensions." International Conference on Learning Representations. 2021.
> 2. Zhuo, Jingwei, et al. "Message passing Stein variational gradient descent." International Conference on Machine Learning. PMLR, 2018.

---

### Official Review · Reviewer_rYZM · 2024-11-06

**Soundness:** 4
**Presentation:** 4
**Contribution:** 3
**Rating:** 8
**Confidence:** 4

**Summary:**

The authors improve Stein variational gradient descent (Liu & Wang, 2016) by extending nonlinear SVGD (Wang & Liu, 2019) by learning a density-based mixture model to approximate the posterior, instead of solely relying on particles (i.e., delta-distributions).
They evaluate their method on a set of (small) scale regression tasks.

**Strengths:**

- The method is a straightforward and effective extension of SVGD/NSVGD
- The paper is well-written and easy to follow and the same goes for the provided codebase

**Weaknesses:**

- The experiments are rather small-scale and limited to regression data sets. Their aim seems to be primarily to demonstrate the relative performance of the proposed approach compared to prior SVGD-related approaches rather than, its absolute performance. In the list of baselines, at least a comparison against an HMC performance on the UCI data sets would have been nice to see how close it can come to it (or improve upon it).
- The paper lacks ablations to evaluate what happens as an underlying BNN gets deeper, i.e., to what extent it can handle the increase in parameters. A deep experiment could be a combination with last-layer BNNs, i.e., learn the mixture not for the whole net, but treat only the penultimate layer in a Bayesian fashion.
- ~~The experiments are limited to regressions with a homoscedastic, known observation noise.~~ What about classification or heteroscedastic regression tasks? _Edit: The claimed fixed noise was an error on my side. See the answer of the authors on this_
- Citing Agarap (2018) in l478 as a reference for ReLUs seems rather odd. In their work, they evaluate the usage of a ReLU in place of a softmax for classification, i.e., nothing related to the current work nor has the ReLU been introduced in that paper.

### Typos
- l233 lacks a second closing bracket

**Questions:**

- Q1: Can the authors speculate on performance as a function of the parameter count, e.g., sticking to BNNs, at which depth/width would the method start to struggle?
- Q2: What are the increased runtime costs compared to compared baselines?

---

> ### Author Response · Authors · 2024-11-19
> **Rebuttal**
>
> ## The experiments are rather small-scale and limited to regression data sets...
>
> You are correct that our experiments are specifically designed to demonstrate how SMI can address variance collapse in SVGD. To achieve this, we use models that are significantly larger than those examined in prior studies on variance collapse in SVGD that appeared in ICML and ICLR [1,2]. The results in this study provide the necessary foundation for further work, notably on kernels and hyperparameter settings that make an application to larger problems possible.
>
> Adding HMC as a baseline is an excellent suggestion. **We will provide results on HMC with NUTS for the UCI datasets and 1D regression tasks.** As expected, NUTS is comparatively slow, but its prediction performance is good. We will include NUTS as a gold standard for comparison in the article and provide the results as an official comment here. We will also include an application to classification (see below).
>
> ## The experiments are limited to regressions with a homoscedastic, known observation noise.
> While it is accurate that our experiments focus on homoscedastic models, the claim that we exclusively use **known** observation noise is incorrect. In our synthetic 1D regression experiments, the observation noise is indeed known. However, in the UCI experiments, we incorporate a prior over the noise, allowing each method to infer a posterior distribution for the noise parameter. Although the noise parameter is shared across all examples—ensuring the models remain homoscedastic—the key distinction is that the observation noise is not predetermined but rather inferred from the data.
>
> ## What about classification or heteroscedastic regression tasks?
> Heteroscedastic regression is commonly handled in BNNs by allowing the network to output the noise level as a function of the inputs. However, in our UCI experiments, we opted to adhere to the standard setup used in the UCI regression benchmark, which involves a homoscedastic noise model. Consequently, we did not employ a BNN architecture designed for heteroscedastic regression in this specific context.
>
> To address the concern about classification, **we will include results for a classification task using MNIST with a 1 and 2-layer BNN.** This addition will demonstrate the applicability of our approach to classification tasks and provide an evaluation beyond regression.
>
> ## l233 lacks a second closing bracket.
>
> Thanks for bringing this to our attention. We have fixed the typo.
>
> ## References
>   1. Ba, Jimmy, et al. "Understanding the variance collapse of SVGD in high dimensions." International Conference on Learning Representations. 2021.
>   2. Zhuo, Jingwei, et al. "Message passing Stein variational gradient descent." International Conference on Machine Learning. PMLR, 2018.

---

> ### Author Response · Authors · 2024-11-19
> **Rebuttal**
>
> ## Q1: Can the authors speculate on performance as a function of the parameter count, e.g., sticking to BNNs, at which depth/width would the method start to struggle?
>
> We are actively working on scaling SMI to larger models, and our current findings indicate that architectural depth, rather than the sheer number of parameters, presents a key challenge. We believe the problems with depth are  related to the issue of non-identifiability [1], which complicates Bayesian inference considerably. We believe these challenges can be addressed by improvements in kernel design, which is beyond the scope of the current contribution.
>
> We currently know that  SMI begins to struggle with a BNN that has three or more layers, such as a fully connected 3-layered MLP with hidden dimension 100 (109,400 parameters) trained on MNIST. In contrast, a convolutional architecture like LeNet-1 (1995 version), which has only around 2,500 parameters but 8 layers, also proves challenging due to its depth. On the other hand, for a one-layer BNN, width does not appear to be a limiting factor for SMI's performance.
> These observations highlight the importance of addressing depth-related challenges to scale SMI to more complex architectures.
>
> ## The paper lacks ablations to evaluate what happens as an underlying BNN gets deeper...
>
> We appreciate the reviewer's suggestion regarding ablations to evaluate SMI's scalability to deeper BNNs. Scaling SMI to large models is indeed an important research direction. However, understanding how SMI scales to large models and more complex architectures is a substantial undertaking that goes beyond the scope of this work. This paper is intended as foundational research, focusing on demonstrating SMI's ability to address the critical issue of variance collapse in SVGD.
>
> By resolving this core limitation, we establish a strong basis for further exploration, including scaling to deeper networks and incorporating last-layer BNNs. While we acknowledge the importance of such extensions, we believe that the contributions of this work should primarily be evaluated on the successful resolution of variance collapse and its implications for Bayesian inference quality. We view this work as a stepping stone, and future research will build upon these findings to address the challenges of scaling SMI to larger models.
>
> ## Q2: What are the increased runtime costs compared to compared baselines?
>
> **To address this, we will provide per-step timings and commentary in the appendix for clarity and reproducibility.** Below, we reproduce the **per-step** average inference time [sec/step] on the UCI datasets for a range of methods, including SMI.
> On UCI datasets, SMI exhibits slower inference compared to the VI-based baselines. A portion of this overhead arises from JIT compilation, which we believe can be reduced by optimizations in future releases of SMI.
>
> | Dataset/Method |   SMI  |  SVGD  |  ASVGD |   OVI  |   MAP  |
> |----------------|:------:|:------:|:------:|:------:|:------:|
> | Boston         | 0.0014 | 0.0003 | 0.0003 | 0.0002 | 0.0001 |
> | Concrete       | 0.0015 | 0.0004 | 0.0003 | 0.0002 | 0.0001 |
> | Energy         | 0.0017 | 0.0003 | 0.0003 | 0.0002 | 0.0001 |
> | Kin8nm         | 0.0192 | 0.0004 | 0.0004 | 0.0003 | 0.0002 |
> | Naval          | 0.0103 | 0.0004 | 0.0004 | 0.0004 | 0.0002 |
> | Power          | 0.0079 | 0.0004 | 0.0004 | 0.0004 | 0.0002 |
> | Protein        | 0.0468 | 0.0011 | 0.0008 | 0.0004 | 0.0003 |
> | Wine           | 0.0093 | 0.0003 | 0.0003 | 0.0003 | 0.0001 |
> | Yacht          | 0.0059 | 0.0003 | 0.0003 | 0.0003 | 0.0001 |
>
> When considering the recovery point experiment (Table 2), SMI demonstrates significantly improved runtime efficiency. On the mid-sized network, SMI achieves inference times 6x faster than SVGD. This observation suggests that while VI methods excel in runtime on UCI datasets, SMI provides a better trade-off when factoring in performance gains. Thus, in contexts where accuracy and robustness are critical, SMI is preferable despite its higher initial runtime cost.
>
> We believe these observations underline the versatility of SMI, and we aim to address current runtime bottlenecks in future updates.
>
> ## Citing Agarap (2018) in l478 as a reference for ReLUs seems rather odd...
>
> Thank you for bringing this mistake to our attention. The reference was supposed to refer to the paper introducing ReLUs as an activation function in Deep Learning. As far as we know, this is the correct reference:
>
> Fukushima, Kunihiko. "Neocognitron: A self-organizing neural network model for a mechanism of pattern recognition unaffected by shift in position." Biological cybernetics 36.4 (1980): 193-202.
>
> ## References
> 1. Roy, Hrittik, et al. "Reparameterization invariance in approximate Bayesian inference." NeurIPS. 2024.

---

> > ### Comment · Reviewer_rYZM · 2024-11-19
> >
> > Thank you for these detailed answers.
> >
> > And sorry for the claim about the fixed noise. I just checked my notes on the review and don't find a source in the paper as to why I claimed that. (I probably extrapolated from the 1d example.)

---

### Author Response · Authors · 2024-11-21
**MNIST results**

The tables below summarize the performance of 1-layer and 2-layer Bayesian Neural Networks (BNNs) on the MNIST dataset, evaluated across several metrics: confidence (Conf), negative log-likelihood (NLL), accuracy (Acc), Brier score (Brier), expected calibration error (ECE), and maximum calibration error (MCE).

For the 1-layer BNN, SMI outperforms other methods across all metrics except ECE and MCE. Given the robustness of the Brier score compared to ECE and MCE—which are sensitive to the number of bins (100 bins were used in this evaluation)—SMI is considered the best-calibrated method. When evaluating all metrics collectively, SMI stands out as the preferred approach.

For the 2-layer BNN, SMI again outperforms other methods on most metrics, except for the Brier score, which is on par with MAP. We regarded SMI and MAP as the best-calibrated methods for the same reasons outlined earlier. Overall, SMI remains the preferred approach when considering all metrics together.

We exclude HMC with NUTS from the MNIST analysis because HMC does not support subsampling, rendering inference computationally infeasible given our, or any conventional, hardware constraints.

## 1-layer BNN table

| Method | Conf ($\uparrow$)  	| NLL ($\downarrow$) 	| Acc ($\uparrow$)   	| Brier ($\downarrow$)   | ECE ($\downarrow$) 	| MCE ($\downarrow$) 	|
|:------:|:----------------------:|:----------------------:|:----------------------:|:----------------------:|:----------------------:|:----------------------:|
| ASVGD  | $0.972 \pm 0.002$  	| $0.053 \pm 0.004$  	| $0.949 \pm 0.003$  	| $0.074 \pm 0.005$  	| $0.135 \pm 0.007$  	| $0.634 \pm 0.024$  	|
| MAP	| $0.973 \pm 0.001$  	| $0.050 \pm 0.002$  	| $0.952 \pm 0.001$  	| $0.068 \pm 0.000$  	| $0.133 \pm 0.000$  	| $\bf{0.574 \pm 0.000}$ |
| OVI	| $0.921 \pm 0.006$  	| $0.158 \pm 0.012$  	| $0.908 \pm 0.006$  	| $0.106 \pm 0.007$  	| $\bf{0.085 \pm 0.010}$ | $0.630 \pm 0.136$  	|
| SMI	| $\bf{0.979 \pm 0.001}$ | $\bf{0.039 \pm 0.003}$ | $\bf{0.957 \pm 0.003}$ | $\bf{0.065 \pm 0.005}$ | $0.148 \pm 0.012$  	| $0.631 \pm 0.047$  	|
| SVGD   | $0.972 \pm 0.003$  	| $0.054 \pm 0.006$  	| $0.949 \pm 0.004$  	| $0.074 \pm 0.007$  	| $0.139 \pm 0.014$  	| $0.653 \pm 0.048$  	|

## 2-layer BNN table
Method | Conf ($\uparrow$)  	| NLL ($\downarrow$) 	| Acc ($\uparrow$)   	| Brier ($\downarrow$)   | ECE ($\downarrow$) 	| MCE ($\downarrow$)
:------:|:----------------------:|:----------------------:|:----------------------:|:----------------------:|:----------------------:|:----------------------:
 ASVGD  | $0.956 \pm 0.004$  	| $0.104 \pm 0.011$  	| $0.936 \pm 0.004$  	| $0.083 \pm 0.003$  	| $0.132 \pm 0.011$  	| $0.651 \pm 0.075$
 MAP	| $0.976 \pm 0.001$ | $0.044 \pm 0.003$ | $0.955 \pm 0.001$ | $\bf{0.066 \pm 0.000}$ | $0.126 \pm 0.000$ | $\bf{0.614 \pm 0.000}$
 OVI	| $0.913 \pm 0.005$  	| $0.182 \pm 0.012$  	| $0.899 \pm 0.005$  	| $0.116 \pm 0.005$ | $\bf{0.084 \pm 0.009}$ | $0.652 \pm 0.133$
 SMI	| $\bf{0.979 \pm 0.002}$ | $\bf{0.042 \pm 0.005}$ | $\bf{0.956 \pm 0.002}$ | $\bf{0.067 \pm 0.003}$ | $0.150 \pm 0.014$  	| $0.653 \pm 0.057$
 SVGD   | $0.960 \pm 0.004$  	| $0.091 \pm 0.011$  	| $0.940 \pm 0.002$  	| $0.081 \pm 0.004$  	| $0.135 \pm 0.013$  	| $0.649 \pm 0.044$

---

### Author Response · Authors · 2024-11-27
**Rebuttal Revisions**

The reviewers' feedback and suggestions have directly contributed to improving the evaluation of our method as well as enhancing the overall clarity and quality of the work. Your comments and critiques have been valuable in refining both our methodology and presentation. Thank you.

We have updated the article with the following:
- Added discussion of ADVI for mixture models to Related Works
- Added experiments showing that the Ba et al. (2021) sampling algorithm is biased to the appendix.
- Added HMC with NUTS results to UCI benchmark and 1D regression.
- Added MNIST results to the main article.
- Added forces discussion to the Appendix.
- Added UCI timings to the Appendix.
- Moved recovery point to the Appendix to make space for the MNIST experiment.

---

### Meta-Review · Area_Chair_9yPu · 2024-12-16

**Metareview:**

This study addresses the variance collapse issue in Stein variational gradient descent (SVGD) by proposing a novel method called Stein mixture inference (SMI), which combines existing Nonlinear SVGD with mixture models. The key property of this method is its ability to retain the nonparametric sample-based characteristic of SVGD, while simultaneously maximizing the evidence lower bound (ELBO), as commonly done in variational inference. This connection allows the proposed method to be interpreted within the framework of variational inference. However, as noted by multiple reviewers, the experiments presented in the paper are limited, and it is unclear how the maximization of ELBO contributes to resolving the variance collapse issue. Despite these weaknesses, the idea of linking SVGD with ELBO maximization is innovative and valuable to the community. With appropriate revisions addressing these weaknesses, the study would be considered suitable for acceptance.

**Additional Comments On Reviewer Discussion:**

All reviewers pointed out the lack of numerical experiments in the study since the original paper only included the small scale problems, the regression tasks, and insufficient insights gained from toy data regarding how variance collapse is resolved, and the lack of comparisons with other methods. Many of these issues have been addressed through additional experiments provided by the authors.
Furthermore, reviewers HpKW and dciU claimed the lack of theoretical insights explaining why the proposed method contributes to resolving variance collapse. The authors, however, have proposed addressing these theoretical aspects in future research.

---

### Decision · Program_Chairs · 2025-01-22

Accept (Poster)